# Accurate Estimation of Mutual Information in High Dimensional Data

## Abstract

Mutual information (MI) is a fundamental measure of statistical dependence between two variables, yet accurate estimation from finite data remains notoriously difficult. No estimator is universally reliable, and common approaches fail in the high-dimensional, undersampled regimes typical of modern experiments. Recent machine learning–based estimators show promise, but their accuracy depends sensitively on dataset size, structure, and hyperparameters, with no accepted tests to detect failures. We close these gaps through a systematic evaluation of classical and neural MI estimators across standard benchmarks and new synthetic datasets tailored to challenging high-dimensional, undersampled regimes. We contribute: (i) a practical protocol for reliable MI estimation with explicit checks for statistical consistency; (ii) confidence intervals (error bars around estimates) that existing neural MI estimator do not provide; and (iii) a new class of probabilistic critics designed for high-dimensional, high-information settings. We demonstrate the effectiveness of our protocol with computational experiments, showing that it consistently matches or surpasses existing methods while uniquely quantifying its own reliability. We show that reliable MI estimation is sometimes achievable even in severely undersampled, high-dimensional datasets, provided they admit accurate low-dimensional representations. This broadens the scope of applicability of neural MI estimators and clarifies when such estimators can be trusted.

## 1 Introduction

Mutual information (MI) is a fundamental measure of statistical dependence between two variables (Shannon, 1948). It captures both linear and nonlinear associations, is invariant under reparameterizations, and is zero if and only if the variables are statistically independent. This makes MI a key tool across disciplines, from neuroscience to computer vision (Viola & Wells III, 1997). In systems neuroscience, MI estimation plays an important role in understanding neural coding, analyzing spike trains in single neurons and neural populations, and studying patterns of information transfer across brain areas and behaviors (Tang et al., 2014; Palmer et al., 2015; Panzeri et al., 2001; Strong et al., 1998; Pica et al., 2017; Runyan et al., 2017; Pascual et al., 2024). Similarly, in brain imaging, MI quantifies functional connectivity between brain regions, illuminating effects of neurological disorders (Ince et al., 2017; Hlinka et al., 2011; Li, 2022). MI has also proven useful in domains like protein sequence alignment and contact prediction (Marks et al., 2011; Lu et al., 2020; Sgarbossa et al., 2023), and in the inference of gene regulatory networks Margolin et al. (2006). In computer vision, state-of-the-art models, such as CLIP (Radford et al., 2021), utilize loss functions related to MI to align visual and textual representations. Similarly, self-supervised learning frameworks such as Barlow Twins (Zbontar et al., 2021) can be interpreted as mutual information-based objectives designed to enforce cross-modal correspondences (Abdelaleem et al., 2025).

For continuous variables $X$ and $Y$, MI is $I(X;Y) = \int dx\, dy\, p(x,y) \log_2 \frac{p(x,y)}{p(x)p(y)}$ (measured in $bits$[1]), where $x$ and $y$ are specific values of the variables, $p(\cdot)$ are the corresponding probability densities, and the integration is over the domain of the variables. (For discrete $X$ and $Y$, sums over distributions are used instead.) Since MI is a nonlinear function of $p(x,y)$, substituting an

---

[1]Most estimators are naturally expressed in *nats*, where log is the natural logarithm. Thus in all *derivations* in this paper $\log = \ln$. However, when reporting the MI values, we convert to *bits* for clarity, $\log = \log_2$.

unbiased estimate of $p$ into the definition of MI results in a biased estimate of MI. Typically, the bias of estimators is a more serious problem than their variance, particularly for continuous variables, because MI is invariant under reparameterization, while it is impossible to construct an estimator that is covariant under all reparameterizations (Holy & Nemenman, 2002). Traditional methods, such as histogram-based, k-nearest neighbors (kNN), box-counting, or kernel-based (Kraskov et al., 2004; Gao et al., 2015; Ross, 2014; Amir Haeri & Ebadzadeh, 2014; Khan et al., 2007; Steuer et al., 2002; Daub et al., 2004; Trappenberg et al., 2005; Fransens et al., 2004), struggle to reduce the bias for high-dimensional data since they require the number of samples that grows exponentially with the dimensionality (Walters-Williams & Li, 2009; Khan et al., 2007; Gao et al., 2015; Czyz et al., 2023).

Recent advances in machine learning resulted in neural network (NN)-based estimators for MI, which aim to circumvent the limitations of traditional methods. These estimators frame MI estimation as optimization over a family of functions (Barber & Agakov, 2003; 2004; Nguyen et al., 2010; Poole et al., 2019; Song & Ermon, 2019; van den Oord et al., 2018). In principle, they can work even for very high-dimensional data that are out of reach for traditional methods. For instance, they supposedly compute MI between images, with the dimensionality of thousands. However, the practical accuracy of these methods remains unclear. First, most of them have been tested primarily on synthetic data with simple dependence structures and unrealistically large datasets. Second, since universally good MI estimation without smoothness assumptions on the underlying distribution is impossible (see, e.g., Paninski (2003); Kandasamy et al. (2015)), internal consistency checks are essential to signal whether the output can be trusted. Such checks are not widely adopted. Third, NN estimators depend on hyperparameters, such as criteria for stopping training, and optimal parameter choices are unclear.

Here, we systematically address these gaps. We argue that successful neural MI estimation requires: (i) even high-dimensional data having a *low-dimensional latent structure*; (ii) the critic *sufficiently expressive* (e.g., embedding dimension and nonlinearity) to capture it; and (iii) sufficient data to *resolve statistical dependencies in the latent* (not full data) space. Our **specific contributions** include: (1) developing a practical protocol for MI estimation, which includes an early-stopping heuristic to minimize estimation bias, internal bias checks, an estimation of error bars (confidence intervals)—the latter is particularly rare among MI estimators; (2) introducing *probabilistic critics*, which improve estimation for large MI values; (3) *benchmarking* our approach against other estimators on synthetic and real-world data, demonstrating when and why MI estimation succeeds.

## 2 BACKGROUND

**Traditional MI Estimation.** Estimating mutual information (MI) from finite data is notoriously challenging, especially for high-dimensional continuous variables (Paninski, 2003). To see this, consider a simple argument: suppose each component of $X$ and $Y$ lies within a bounded range of size $A$, and the joint density $p(x, y)$ is smooth, with its smallest feature on the scale of $a$. Then accurate estimation requires $N \gg (A/a)^{K_{\text{tot}}}$ samples, where $K_{\text{tot}} = K_X + K_Y$ is the total dimensionality of $X$ and $Y$. If $A/a > 1$, the required sample size is exponential in $K_{\text{tot}}$, illustrating the classic curse of dimensionality. The situation is even worse when the coordinate system in which $p$ is smooth is unknown (Holy & Nemenman, 2002), or when the variables are unbounded. Thus, while many methods have been developed to estimate MI, they often break down beyond $K_{\text{tot}} \sim 10$ dimensions Holmes & Nemenman (2019) (cf. Appx. Fig. 9). In contrast, most modern datasets are high-dimensional, e.g., images with thousands of pixels, or neural recordings from thousands of units.

**Neural Network-Based Estimators.** The struggle against the curse of dimensionality has motivated neural network (NN)-based approaches since deep NNs can capture complex nonlinear dependencies in high-dimensional data (LeCun et al., 2015). Neural variational methods have become particularly influential for MI estimation (Barber & Agakov, 2003; 2004; Nguyen et al., 2010; Donsker & Varadhan, 1983). Specifically, typically, we do not have access to the full joint distribution $p(x, y)$ or the marginals $p(x)$ and $p(y)$, but we can draw samples from them. Variational estimators leverage this by reformulating MI in terms of a Kullback-Leibler (KL) divergence:

$$I(X; Y) = D_{\text{KL}}\left(p(x, y) \parallel p(x)p(y)\right) = \mathbb{E}_{p(x)}\left[D_{\text{KL}}\left(p(y|x) \parallel p(y)\right)\right]. \tag{1}$$

Then using, the Donsker–Varadhan (DV) representation of the first KL divergence in Eq. (1) (Donsker & Varadhan, 1983), $D_{\text{KL}}(P \parallel Q) \geq \max_T \left\{\mathbb{E}_P[T] - \log \mathbb{E}_Q[e^T]\right\}$, learning the *critic* function $T(x, y)$ via a NN, and replacing expectations with sample averages, one obtains the MINE estimator

(Belghazi et al., 2018; Poole et al., 2019) (Appx. Eq. 7). However, MINE suffers from high variance, and its estimate is not a strict lower bound on MI when the normalization term is approximated by Monte Carlo sampling (Poole et al., 2019). One addresses this by clipping the critic to the range $\pm\tau$, yielding the SMILE estimator (Song & Ermon, 2019) (Appx. Eq. 8). Alternatively, applying the DV representation to the second KL divergence in Eq. (1) instead leads to the InfoNCE estimator (van den Oord et al., 2018), widely used in contrastive learning (Appx. Eq. 11). See derivations for all the methods in Appx. A.1.1. Although early studies employed simple critic networks, the architecture and expressivity of $T(x, y)$ strongly influence estimator performance. We return to this in Sec. 3.

**Limitations of Neural Estimators.** Despite their popularity, it is unclear if existing neural MI estimators are truly accurate, calling their widespread use into question. First, most tests (Czyz et al., 2023) of the estimators to date have been performed on synthetic data with low dimensionality (e.g., $K_X, K_Y \sim 10$), where the true MI is known analytically. However, traditional estimators such as kNN-based methods (Kraskov et al., 2004; Holmes & Nemenman, 2019) already perform well then (Czyz et al., 2023). Unless neural estimators clearly outperform these simpler methods in high dimensions $K \gtrsim 100$), their practical value is limited. Yet, evaluations in this regime remain scarce.

Second, when $X$ and $Y$ are jointly Gaussian, MI can be computed exactly from their correlation matrix. Since correlation matrices can be reliably estimated when $K/N \ll 1$ (Bouchaud et al., 2007; Swain et al., 2025), this provides a natural benchmark. If a neural estimator fails when a linear method succeeds, it is not exploiting all the statistical structure in the data. Nevertheless, such comparisons are rare. As we will demonstrate below, some neural estimations fall short on this metric.

Finally, estimators are often validated in effectively infinite-sample regimes (Poole et al., 2019; Song & Ermon, 2019), with a fresh data batch at every training step. This sidesteps overfitting and gives an overly optimistic view of estimator performance. In real life, sample size is often small ($N \sim K$), and success for infinite-data does not imply practical utility. We will show that unbiased estimation is sometimes possible even in this heavily undersampled regime if data have a simple latent structure.

## 3 A GENERALIZED CRITIC

Neural network-based MI estimators typically rely on a *critic* function $T(x, y)$ that approximates a log-density ratio. In a unified formulation, the critic can be expressed as:

$$T(x, y) = f(g(x), h(y)), \tag{2}$$

where $g : \mathcal{X} \to \mathcal{Z}_X$ and $h : \mathcal{Y} \to \mathcal{Z}_Y$ are embedding functions, and $f : \mathcal{Z}_X \times \mathcal{Z}_Y \to \mathbb{R}$ combines the embeddings into a scalar score. Different choices of $f$, $g$, and $h$ recover many well-known estimators and permit novel architectures tailored to specific tasks. Specifically:

**Joint (Concatenated) Critic:** Setting $g, h$ to the identity maps and letting $f$ be a NN that operates on the concatenated inputs reproduces the joint-critic architecture used in MINE (Belghazi et al., 2018).

**(Deep) Separable Critic:** Choosing $f(g, h) = \langle g, h \rangle$, with $g, h$ being vector-valued embeddings (e.g., via multilayer perceptrons), yields the separable critic of InfoNCE (van den Oord et al., 2018).

Although certain critics are commonly paired with specific objectives, this pairing is not obligatory. For instance, separable critics, typically used with contrastive losses, can be combined with non-contrastive objectives. Likewise, joint critics can be used in contrastive settings.

In this work, we introduce or reformulate additional critic choices:

**Concatenated Quadratic Critic:** We show that if $g$ and $h$ are linear projections (e.g., identity maps) and $f$ is a quadratic form of its concatenated arguments, then MI estimation, denoted here as $I_{\text{CCA}}(X; Y)$, reduces to measuring canonical correlations (Hotelling, 1936) in a shared low-dimensional space (Appx. A.1.2).

**Probabilistic Critic (VSIB Framework):** Rather than using deterministic mappings, $f$, $g$, and $h$ can be stochastic, leading to variational objectives that regularize embedding distributions. We implement this with a loss similar to that introduced by Abdelaleem et al. (2025) (Appx. A.2):

$$L_{\text{EST}-\text{VSIB}} = I^E(X; Z_X) + I^E(Y; Z_Y) - \beta I^D_{\text{EST}}(Z_X; Z_Y), \tag{3}$$

where the $I^E$ terms encourage compressive embeddings, and $I^D$ measures the mutual information between latent variables using a selected estimator EST.

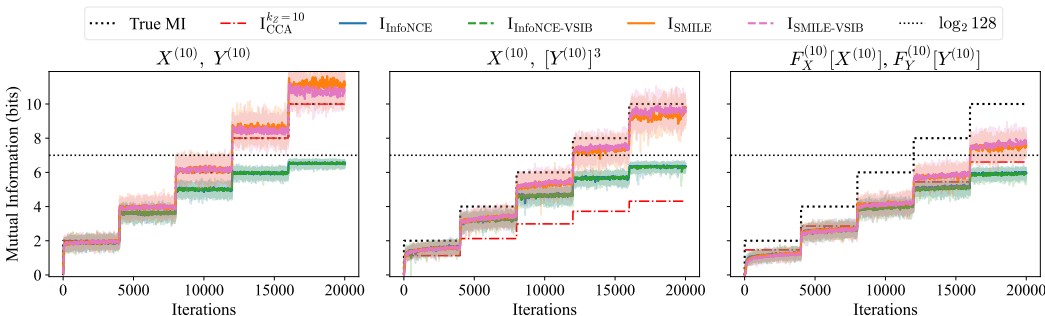

Figure 1: **MI estimators in the low-dimensional, infinite-data regime.** Each panel plots running MI estimates over training iterations for five true MI levels (increasing every 4000 iterations). Each step introduces a fresh batch of 128 samples. We compare the CCA-based estimator (optimal for Gaussian data), InfoNCE, SMILE, and their probabilistic variants (denoted with VSIB). Faint curves show raw estimates; bold curves show smoothed trends (Appx. A.6). Left: For jointly Gaussian $X, Y$, all estimators initially perform well. InfoNCE plateaus at its well-known intrinsic upper bound (van den Oord et al., 2018) $\log(\text{batch size}) \approx 7$ bits, while SMILE begins to overestimate at high MI, indicating overfitting. $I_{\text{CCA}}$ overlaps with ground truth, as expected. Middle: Cubing $Y$ breaks linearity, and $I_{\text{CCA}}$ fails. Nonetheless, InfoNCE behavior is almost unchanged, and SMILE remains reasonably effective with sufficient training (Appx. Fig. 10). Both slightly underestimate at low MI, and for SMILE this is largely offset by its intrinsic positive bias at high MI. Right: Passing $X$ and $Y$ through separate frozen teacher networks (one hidden softplus layer, 1024 units) creates highly nonlinear dependencies. All estimators underestimate MI.

## 4 RESULTS

### 4.1 ESTIMATOR PERFORMANCE IN THE INFINITE DATA REGIME

We first evaluate neural MI estimators in an idealized, effectively infinite-sample setting that eliminates overfitting. Although common in prior work, this regime obscures many real-world challenges. For benchmarking, we use both low- and high-dimensional synthetic data with known ground-truth MI. We adopt the following notation. $X$ and $Y$ are the variables whose MI we estimate; $Z_X$ and $Z_Y$ denote their low-dimensional embeddings by a critic. The observed dimensionalities are $K_X$ and $K_Y$ ($K$ if both are equal), while $K_Z$ is the true latent dimensionality when it differs from $K$. Finally, $k_Z$ is the embedding dimensionality of the critic, which may differ from $K_Z$.

**Low-Dimensional $X$ and $Y$.** We begin with low-dimensional data ($K_X = K_Y = 10$), where both traditional (e.g., kNN) and neural estimators are expected to perform well. We first analyze jointly Gaussian data with a chosen correlation. It is well known that, in this case, MI estimation reduces to the sum of information in each canonical correlation pair (Gelfand & Iaglom, 1959; Kullback, 1997; Huffmann & Mittelbach, 2022), $I_{\text{CCA}} = -\frac{1}{2} \sum_i^{K_Z} \log(1 - \rho_i^2)$, where $\rho_i$ is the canonical correlation. We then increase complexity by (i) cubing $Y$ or (ii) passing both variables through separate fixed teacher networks, thereby introducing diverse nonlinear dependencies.

Figure 1 (left) confirms that InfoNCE, SMILE, and CCA track the true MI on Gaussian data (until InfoNCE saturates (van den Oord et al., 2018)). The data matrices used here are aggregated across all batches seen during training at a given MI level. If the critic is expressive enough, SMILE is expected to show high variance and to overestimate at high MI since it evaluates log-sum-exp in its DV bound, Eq. (7), which is biased due to the Jensen's inequality (Guo et al., 2022; Choi & Lee, 2020). We observe both problems, but neither is strong, at least at low MI.

Cubing $Y$ introduces nonlinear correlations, and $I_{\text{CCA}}$ fails, Fig. 1 (middle). In contrast, neural estimators recover MI after sufficient training (Appx. Fig. 10). Both exhibit a small but significant negative bias at low MI, likely due to the reduced invertibility of the cubic map near $y = 0$. When $X$ and $Y$ are passed through separate frozen teacher networks, Fig. 1 (right), all estimators underestimate MI, although neural methods still outperform the CCA baseline. We again attribute this bias to the reduced invertibility, now due to the softplus saturation in the teacher networks.

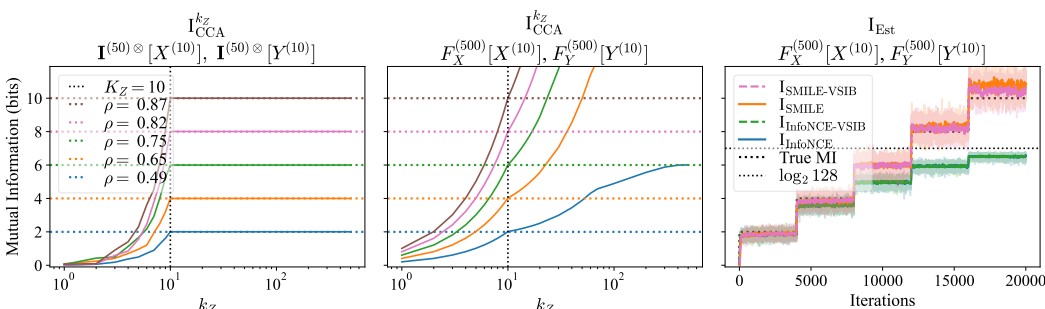

Figure 2: **MI estimators in the high-dimensional, infinite-data regime.** We extend Fig. 1 to $K_X = K_Y = 500$, embedding $K_Z = 10$ latent variables into high-dimensional $X$ and $Y$. Left: A linear transformation (e.g., identity and replication) expands $Z$ to 500-dimensional $X$ and $Y$. $I_{CCA}$ with $k_z \geq K_Z = 10$ dimensions still accurately recovers the ground truth. Middle: A frozen nonlinear teacher network maps $Z$ to 500-dimensional $X$ and $Y$. Unlike the linear case, $I_{CCA}$ fails due to the nonlinearity of the transformation. Increasing $k_Z > K_Z$ detects spurious correlations, inflating MI estimates and illustrating the limitations of linear methods in nonlinear settings. Right: Neural estimators (InfoNCE, SMILE, and their VSIB variants) are applied directly to the full 500-dimensional data. All are accurate across the full range of true MI values, performing even better than in Fig. 1 due to improved invertibility of the nonlinear transformation in high dimensions.

Appx. Fig. 8 surveys other neural estimators and confirms that InfoNCE and SMILE consistently outperform the rest. Therefore, in what follows, we restrict our analysis to these two methods, using a fixed clipping factor $\tau = 5$ for SMILE. Combined, these results validate MI neural estimators—specifically InfoNCE, SMILE, and their VSIB variants—in low dimensions with abundant data. However, they still underestimate MI for strongly nonlinear dependencies, even with effectively unlimited data. Thus, seemingly, there is little reason to prefer them over simpler correlation-based or kNN methods, which already perform well in these scenarios (Kraskov et al., 2004; Holmes & Nemenman, 2019; Czyz et al., 2023).

**High-Dimensional $X, Y$.** To distinguish effects of the observed and latent dimensionality, we increase the former, while fixing the latter. We start again with the jointly Gaussian 10-d $X$ and $Y$ with the known ground truth MI. We then (i) replicate each of the ten components of $X$ and $Y$ 50 times (denoted $X \rightarrow \mathbf{I}^{(50)} \otimes [X]$), and (ii) pass $X$, $Y$ through distinct frozen teacher networks, embedding each into 500 dimensions (denoted $X \rightarrow F_X^{(500)}[X]$, and similarly for $Y$). Both cases result in $K_X = K_Y \equiv K = 500$, while $K_Z = 10$, with the former case having only linear correlations.

Figure 2 shows that, in the linear replication setup, $I_{CCA}$ accurately recovers the true MI when $k_Z \geq K_Z$. In the nonlinear case, the CCA approach breaks down: as the number of detected canonical pairs $k_Z$ increases, the method approximates nonlinear dependencies with an ever larger set of linear projections, inflating the MI estimate with no upper bound. In contrast, neural estimators (InfoNCE, SMILE, and their VSIB variants) recover the ground truth MI when applied directly to the full $K = 500$-dimensional spaces (up to InfoNCE's saturation). Performance even surpasses the low-dimensional case, Fig. 1, because non-invertible, saturated softplus regions in one random embedding can be inverted in others, allowing reconstruction of the full latent manifold.

This highlights two points. First, neural estimators may work well in high-dimensional nonlinear settings, where traditional approaches fail (Holmes & Nemenman, 2019) (Appx. Fig. 9). Second, matching the estimator to the structure of the data (e.g., using $I_{CCA}$ for linear correlations) may result in more computationally and data-efficient estimation (Abdelaleem et al., 2024).

## 4.2 ESTIMATOR PERFORMANCE WITH FINITE DATASETS

For infinite-data, neural estimators receive a fresh data batch at every training step. Yet, in practice, datasets are always finite. The impact of a finite sample size on the estimators remains poorly understood (though see Czyz et al. (2023)); here we address this gap.

**The Stopping Heuristic.** Figure 3 demonstrates overfitting for finite data in the high-dimensional teacher model of Fig. 2 (ground-truth MI of 4 bits). We track $I_{\text{InfoNCE}}$ and $I_{\text{SMILE}}$ on training and held-out sets for $2^8 = 256$ (*undersampled*) and $2^{14} = 16,384$ (*better sampled*) pairs of $K = 500$-dimensional $X$ and $Y$ as training progresses. Because 64 epochs in the undersampled setting expose the network to the same number of examples as a single epoch in the better-sampled case (but with repetition), direct epoch counts are not comparable. All training curves start below the true MI and then rise

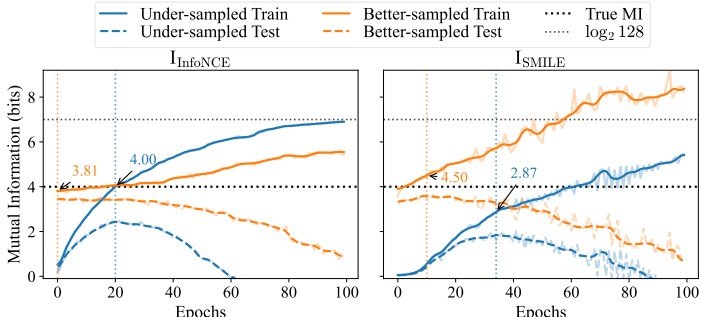

Figure 3: **The stopping heuristic.** We evaluate neural MI estimators for finite-data using the teacher model from Fig. 2, where 10 latent variables carrying 4 bits of MI are embedded in 500-dimensional $X$ and $Y$. We compare two sampling regimes for InfoNCE (left) and SMILE (right): 256 samples (*under-sampled*) and a larger dataset of $2^{14} = 16{,}384$ samples (*better-sampled*). In all cases, the test-set MI initially rises before declining due to overfitting (we do not show the negative values). The **stopping heuristic** selects the epoch with the peak test MI but reports the corresponding training MI. Here the batch size is 128, so that InfoNCE does not saturate.

(InfoNCE until the saturation) as the networks fit finer-scale structure of the critics. Because the training curves show no clear inflection when they surpass the ground-truth, common heuristics (fixed epoch counts or loss plateau) give no reliable stopping signal. In contrast, the test curves grow initially but soon collapse, revealing overfitting.

To devise a better stopping rule, note that the training time effectively sets the smallest scale resolved in the neural approximation of the DV-optimal critic. Early stopping oversmooths and underestimates MI, whereas late stopping undersmooths, thus pushing the training MI high while the test MI falls. This mirrors kernel density estimation (KDE) of MI (Moon et al., 1995). The optimal resolution must depend on data complexity and sample size, so no fixed epoch rule would work.

In KDE MI estimation, one chooses the bandwidth that maximizes held-out likelihood (Margolin et al., 2006). Since the joint-density term dominates test error, this maximizes test MI. Analogously, our stopping heuristic is: track MI on a test batch each epoch, pick the epoch where this test value peaks, and report the corresponding *training MI* as the estimate (see Appx. A.3 for justification). To our knowledge, such rule has not been formalized for neural MI estimation.

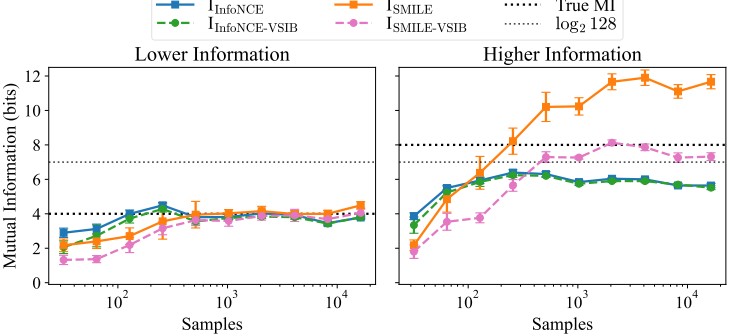

Figure 4: **MI vs. sample size for low and high information.** We compare InfoNCE, SMILE, and VSIB versions with the max-test stopping for different sample sizes. Data from the frozen teacher model (10 latent, 500 data dimensions). All estimators use separable critics, $k_z = 32$. Means $\pm$ s.d. over 10 trials shown. Left: For small MI (4 bits), all estimators recover the ground truth for $10^2 \lesssim N < K = 500$. Right: For high MI (8 bits), contrastive estimators (InfoNCE and InfoNCE $-$ VSIB) saturate near $\log(\text{batch size}) = 7$ bits. SMILE overestimates dramatically as $N$ grows. SMILE $-$ VSIB tracks the ground truth accurately for all $N \gtrsim 10^2$.

**Probabilistic Embeddings Reduce Estimator Bias and Variance.** Figure 4 compares InfoNCE, SMILE, and their probabilistic variants for varying sample sizes and different ground truth MI for the $K = 500$, $K_Z = 10$ random teacher networks model. For small MI (4 bits), all estimators converge to truth when $K_Z \ll N \sim O(10^2) < K$. Thus MI estimation requires good sampling of the latent (but not the data) space. As before, InfoNCE has lower variance than SMILE.

For high true MI (8 bits), all variational bounds and corresponding estimators degrade (Poole et al., 2019). InfoNCE and InfoNCE $-$ VSIB both saturate near $\log(\text{batch size}) = 7$ bits, as expected (van den Oord et al., 2018). Consistent with Figs. 1, 2, SMILE substantially overestimates when large sample size allows overtraining. In contrast, SMILE $-$ VSIB remains accurate and stable, converging to the correct value at $K_Z \ll N \lesssim K_X + K_Y$. This highlights the utility of the new **probabilistic critic family**. By treating the critic output as a distribution (e.g., Gaussian) and regularizing via the encoding terms (Appx. A.2), these critics mitigate the pathological variance and overfitting in SMILE. Their benefit is most pronounced for high MI, where resolving fine-grained data structure is essential.

**Low-dimensional Latent Structure Allows Reliable Estimation.** Our experiments have used data with statistical dependencies in a relatively low-dimensional latent space, $K_Z = 10$. We now ask how the latent dimensionality limits neural MI estimation by simultaneously varying the true latent dimension, $K_Z$, and the critic's embedding dimension, $k_Z$, while keeping the data dimension $K = 500$ large and fixed. We consider a low-dimensional latent setting, $K_Z = 1$, a moderate regime, $1 \ll K_Z = 100 \ll K$, and a fully high-dimensional latent regime, $1 \ll K_Z = K = 500$. With the ground-truth MI at 4 bits (where all estimators can work, Fig. 4), each canonical pair contributes 0.4, 0.04, and 0.008 bits (equivalent correlation $\rho \approx 0.65, 0.23$, and $0.11$, respectively), before they are nonlinearly mixed by the teacher networks. We expect that, if $k_Z \geq K_Z$, the critics can recover dependencies in the low-dimensional case (similar to Abdelaleem et al. (2024)). However, as $K_Z$ increases, the critic must disentangle an ever-growing number of weak interactions, and its MI estimate will deteriorate at a fixed $N$.

Figure 5 shows results for InfoNCE; not shown estimators behave similarly. When the latent space is small ($K_Z = 10$, left), a critic with $k_Z \geq K_Z$ captures all dependencies once $N \gg K_Z$, and the estimate approaches the truth. With a moderate latent dimension ($K_Z = 100$, center), setting $k_Z < K_Z$ still fails outright, while $k_Z \geq K_Z$ retrieves only part of the information: even $N \sim 10^5$ samples result in $< 4$ bits. If the latent and observed dimensions coincide ($K_Z = K = 500$, right), neither an over-sized critic nor $N \sim 10^5$ samples is sufficient to recover information. The sample requirement in affected in two ways: for larger effective dimension, generic methods demand more data (Khan et al., 2007; Gao et al., 2015; Czyz et al., 2023), and contribution of each dimension to the total information is smaller and harder to detect. Surprisingly, the number of samples needed to estimate nonzero MI matches signal detection limits in spiked covariance models from random matrix theory (Baik et al., 2005; Potters & Bouchaud, 2020) in the *latent* and *not* the data space (Appx. A.5).

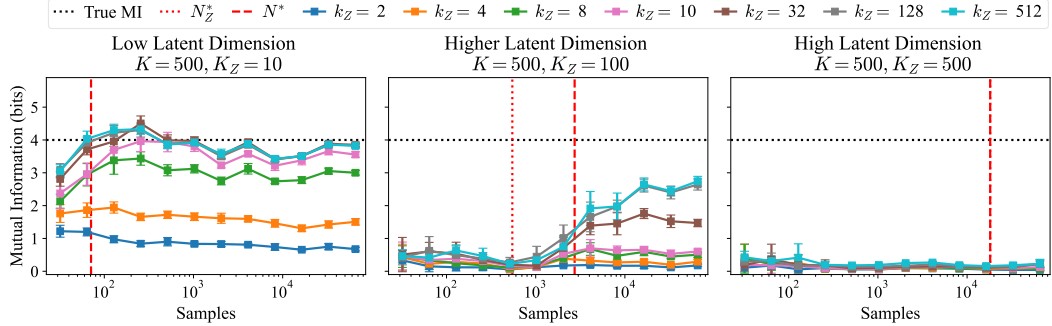

Figure 5: **Effect of latent and critic dimensionality on InfoNCE.** Curves show mean $\pm$ s.d. over ten runs. Panel represents $K = 500$-dimensional data generated by teacher networks with latent dimensionality $K_Z = 10, 100$, and $500$ (left to right). The true MI is 4 bits throughout. A sufficiently expressive critic ($k_Z \geq K_Z$) is required to recover all the information, yet the estimate approaches 4 bits only in the low-dimensional latent case ($K_Z \ll K$) when sample size satisfies $N \gg K_Z$. For larger latent spaces, the estimate remains far below the target even with large $N$. Vertical lines mark $N$ needed for detection of nonzero MI using Gaussian random matrix models in the latent space ($N_Z^*$) and in the full space ($N^*$), cf. Appx. A.5 ($N_Z^* \approx 1$ not shown in the left panel; and $N_Z^* = N^*$ in the right panel). Since a nonzero estimate emerges at $N > N_Z^*$, but $N < N^*$ if $K_Z \ll K$, sampling of the latent space (not the full data space) governs the estimation even in the non-Gaussian setting.

Thus, accurate MI estimation requires that: (i) the dependencies can be represented in a low-dimensional latent space, (ii) the critic is expressive enough to model that space, and (iii) the dataset is sufficient to sample that space, with the size rising quadratically with latent dimension (Appx. A.5). Therefore, in high dimensions, compressive embeddings are not just advantageous but essential. Using methods such as in Kandasamy et al. (2015), it should be possible to formalize these results and show that, as long as sufficiently smooth embedding can be constructed, reliable estimation with fast convergence with the data set size should be possible directly in the embedding space.

### 4.3 Practical Guide to MI Estimation

Because no estimator is unbiased for all distributions (Paninski, 2003), sample-size dependent bias is the dominant practical problem. The remedy is empirical: test how the estimate changes when the sample size varies (Strong et al., 1998; Nemenman et al., 2004; Tang et al., 2014). The *best practice* randomly partitions the data into $\gamma = 1, 2, 3, \ldots$ equal subsets and computes MI for each subset, $I_\mu(\gamma)$, $\mu = 1, \ldots, \gamma$. One checks if $I_\mu(\gamma)$ is statistically stable as $\gamma$ varies (Holmes & Nemenman, 2019), or if it extrapolates linearly to the hypothetical $\gamma = 0$ limit (effectively, infinite data)

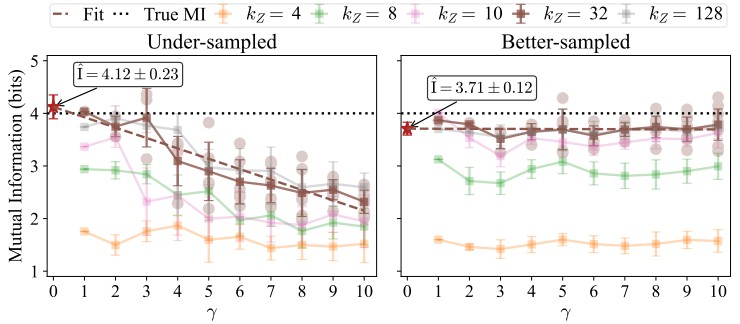

Figure 6: **Workflow for MI estimation.** We apply InfoNCE with a separable critic to data from a random-teacher model ($K_X = K_Y = 500$, $K_Z = 10$, true MI 4 bits). (left) Undersampled, $N = 256$. (right) Well-sampled, $N = 2^{14} = 16{,}384$. For each panel, MI is computed on $\gamma$ equal random, non-overlapping subsets (shown as circles for $k_Z = 32$); error bars are $\pm$ s.d. over subsets. Estimates plateau at $k_Z = 32$ (used in final fits), indicating sufficient expressivity. A linear fit to $\gamma \to 0$ (infinite-data limit) gives the reported MI value. The extrapolation removes sample-size dependent bias in the undersampled case; no such bias is visible in the well-sampled regime.

(Strong et al., 1998). Large curvature in $I_\mu(\gamma)$ vs $\gamma$ signals that the estimator is not in its asymptotic regime and is unreliable. In addition, the scatter of $I_\mu(\gamma)$ at fixed $\gamma$ provides an empirical variance that can be extrapolated to $\gamma = 1$ (all data) or $\gamma = 0$ (infinite data) (Holmes & Nemenman, 2019).

We combine these best practices with our analysis into the following guidelines for neural MI estimation (see Appx. A.4 for details):

1. Select an estimator/critic (EST) suited to the expected MI range and data type—e.g., InfoNCE for modest MI, SMILE–VSIB for high MI, or a separable VSIB critic when probabilistic embeddings are desirable.

2. Choose a critic network architecture that matches the data: a multilayer perceptron for generic data, a convolutional network for images, a transformer for sequences, etc.

3. For $\gamma = [1, 10]$ compute $I_{\text{EST},\mu}^{k_Z}(\gamma)$ starting with $k_Z = 1$ using the max-test early-stopping.

4. Estimate $\bar{I}_{\text{EST}}^{k_Z}(\gamma) \pm \sigma^{k_Z}(\gamma)$ as sample mean and standard deviation over subsets at fixed $\gamma$.

5. Increase $k_Z$ and repeat steps 3–4 until $\bar{I}_{\text{EST}}^{k_Z}(\gamma)$ vs $k_Z$ no longer rises significantly. The smallest dimension that reaches this plateau is $k_Z^*$; modestly over-estimating $k_Z^*$ is safe.

6. If $I_{\text{EST},\mu}^{k_Z^*}(\gamma)$ vs $\gamma$ is approximately linear, extrapolate to $\gamma \to 0$ (details in Appx. A.4).

7. Report the extrapolated value $\hat{I}$ as the MI estimate together with its prediction interval $\Delta I$. If linear extrapolation is impossible, report failure to estimate.

Figure 6 illustrates the workflow on data from a random teacher model with $K = 500$, $K_Z = 10$ and true MI of 4 bits. The left and the right panels show the undersampled ($N = 256$) and the well-sampled ($N = 2^{14}$) cases, respectively. On the left, using $k_Z^* = 32$, we can reliably extrapolate $I_{\text{InfoNCE}}$ to $N \to \infty$ limit ($\gamma = 0$) via subsampling. On the right, the estimator is already stable at this $N$; its slight downward bias matches Fig. 5 (see Discussion). We stress how striking this result is: in a $500 \times 500$-dimensional, highly nonlinear setting (but with a 10-d latent structure), we obtain a near-perfect MI estimate, complete with accurate error bars, from just 256 samples!

**Evaluations I: Benchmarking.** We illustrate advantages of our pipeline in a subset of standard benchmarks (Czyz et al., 2023) in Tbl. 1. Our approach consistently estimates MI better than other methods, while also providing error bars. Additional comprehensive benchmarks in Appx. A.7.3.

Table 1: Comparison to selected benchmarks of Czyz et al. (2023) (results in *nats* to match the reference).

| Task | True MI | simple InfoNCE | Ours |
|------|---------|----------------|------|
| Asinh @ Student-t $3 \times 3$ | 0.29 | 0.20 | $0.26 \pm 0.03$ |
| Multinormal $25 \times 25$ | 1.29 | 1.20 | $1.26 \pm 0.03$ |
| Multinormal $50 \times 50$ | 1.62 | 1.40 | $1.60 \pm 0.03$ |
| Normal CDF @ Multinormal $25 \times 25$ | 1.02 | 0.80 | $0.94 \pm 0.12$ |
| Spiral @ Multinormal $25 \times 25$ | 1.02 | 0.70 | $0.97 \pm 0.24$ |
| Student-t $3 \times 3$ | 0.29 | 0.20 | $0.26 \pm 0.04$ |
| Student-t $3 \times 3$ | 0.18 | 0.10 | $0.18 \pm 0.03$ |

**Evaluations II: A Real-World Example, Noisy MNIST.** We apply our pipeline to a realistic *noisy MNIST* dataset (LeCun et al., 1998; Wang et al., 2015; 2016; Abdelaleem et al., 2025). Each sample consists of two $28 \times 28 = 784$ dimensional views: $X$ is a randomly rotated and scaled image of a digit, and $Y$ is another random digit with the same label overlaid with Perlin noise (Appx. Fig. 11). The only shared information is the digit label (10 classes), giving the ground truth MI of $\log_2 10 \approx 3.3$ bits. The data dimension is $K = 784$, while the latent dimension is low (but unknown) because only ten classes matter. Random matrix analysis, Appx. A.5, suggests that the nonzero MI detection thresholds are $N_Z^* \approx 17$ (for $K_Z = 10$) or $N^* \approx 2700$. Numerical experiments show that $N \sim 512$ is sufficient for detection, so that, again, good sampling in the latent (not data) space is sufficient. Here, however, we need *accurate* estimation of MI, rather tl

For this, we show in Fig. 7 that, with $N = 2^{14} = 16,384$ samples, the estimator works: reliable linear extrapolation to $\gamma = 0$ yields $\hat{I} = 3.13 \pm 0.12$ within two error bars of true MI. A slight kink at $\gamma = 4$ indicates that the estimator will underestimate for $N \lesssim 4 \times 10^3$. That the estimator requires more data than in Fig. 6 is because of the clustered, non-smooth structure of the data in the latent space. Thus reliable estimation on noisy-MNIST requires several thousand examples but not the hundreds of thousands traditional methods would need in 784 dimensions!

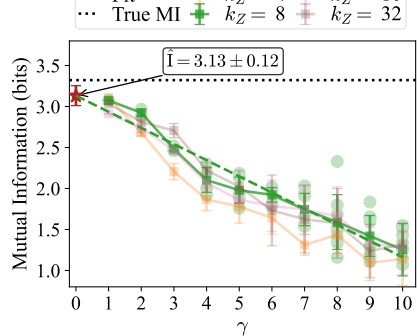

Figure 7: **MI estimation on noisy MNIST.** From $16,384$ samples, our approach reliably estimates MI in this $K_X = K_Y = 784$-dimensional dataset.

## 5 DISCUSSION

Accurate MI estimation for high-dimensional data remains difficult because no estimator is uniformly unbiased. Neural estimators offer flexibility, but most studies test them in well-sampled, linear settings that hide their limitations. We show that neural estimators become reliable when: the statistical dependence lies in a low-dimensional latent space; the critic is expressive enough to capture that space; and the dataset is large enough to resolve it. Under these conditions, three improvements make neural MI estimation practical. First, a "max-test" early-stopping rule prevents the runaway growth common in DV objectives. Second, using the MINE family critics within a probabilistic VSIB wrapper regularizes them at high information values. Third, a subsampling-and-extrapolation workflow detects bias, chooses the right critic dimension, and estimates error bars. Although we illustrated this with InfoNCE and specific network architectures, our analysis can incorporate any DV-based estimator.

Using these ingredients we obtained near-exact MI estimates, with error bars, in regimes that defeat most methods (Appx. Fig. 9), e.g., 784-dimensional images with only $\sim 10^4$ samples. Across all experiments, our procedure never significantly overshot the ground truth, which is crucial to not produce false positives in typical scientific applications (Strong et al., 1998; Margolin et al., 2006; Tang et al., 2014), whereas modest underestimation vanishes as $N$ grows and finer structure is learned. This is also why the small downward bias sometimes visible at large $N$ (cf. Fig. 6 (right)), which is an artifact of a fixed batch size unable to explore fine-scale data features, is not a major concern.

High-dimensional MI estimation will always require careful diagnostics and a measure of skepticism, but our approach turns neural estimators into practical tools for regimes that were previously inaccessible, $N \lesssim K$, potentially strongly impacting how MI estimation is used in scientific research.

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

## A  TECHNICAL APPENDICES AND SUPPLEMENTARY MATERIAL

### A.1  DERIVING DIFFERENT ESTIMATORS

#### A.1.1  INFONCE AND SMILE

Two widely used neural estimators of mutual information—InfoNCE and SMILE—can be derived from the Donsker–Varadhan (DV) representation (Donsker & Varadhan, 1983) of the Kullback–Leibler (KL) divergence.

Mutual information (MI) between random variables $X$ and $Y$ can be expressed as the KL divergence between their joint distribution and the product of marginals:

$$I(X;Y) = D_{\mathrm{KL}}(p(x,y)\|p(x)p(y)) = \mathbb{E}_{p(x,y)}\left[\log \frac{p(x,y)}{p(x)p(y)}\right]. \tag{4}$$

Alternatively, MI can be factorized using the conditional distribution:

$$I(X;Y) = \mathbb{E}_{p(x)}\left[D_{\mathrm{KL}}(p(y|x)\|p(y))\right] = \mathbb{E}_{p(x)}\left[\mathbb{E}_{p(y|x)}\left[\log \frac{p(y|x)}{p(y)}\right]\right]. \tag{5}$$

The DV representation provides a lower bound on KL divergence between two distributions $P$ and $Q$:

$$D_{\mathrm{KL}}(P\|Q) \geq \sup_{T}\left[\mathbb{E}_P[T] - \log \mathbb{E}_Q[e^T]\right], \tag{6}$$

where the supremum is over all measurable functions $T$ such that $\mathbb{E}_Q[e^T] < \infty$. Equality is achieved when $T^* = \log \frac{dP}{dQ} + c$ for any constant $c$.

Applying the DV representation to Eq. equation 4, one obtains:

$$I(X;Y) \geq I_{\mathrm{MINE}}(X;Y) := \max_{T}\left[\mathbb{E}_{p(x,y)}[T(x,y)] - \log \mathbb{E}_{p(x)p(y)}\left[e^{T(x,y)}\right]\right], \tag{7}$$

where $T(x,y)$ is a learned critic function approximating the log-density ratio. This is the MINE estimator (Belghazi et al., 2018; Poole et al., 2019), which exhibits large variance empirically, specifically because of the second term.

To stabilize this estimator, SMILE (Song & Ermon, 2019) clips the critic before exponentiation to reduce the influence of outliers on the normalization term:

$$I(X;Y) \geq I_{\mathrm{SMILE}}(X;Y) := \max_{T}\left[\mathbb{E}_{p(x,y)}[T(x,y)] - \log \mathbb{E}_{p(x)p(y)}\left[e^{\mathrm{clip}(T(x,y),-\tau,\tau)}\right]\right], \tag{8}$$

where $\mathrm{clip}(z,-\tau,\tau) = \min(\max(z,-\tau),\tau)$, and $\tau > 0$ effectively controls the bias–variance trade-off.

Using the conditional factorization of MI in Eq. equation 5, one can again apply the DV representation, now to $D_{\mathrm{KL}}(p(y|x)\|p(y))$:

$$D_{\mathrm{KL}}(p(y|x)\|p(y)) \geq \sup_{T_x}\left[\mathbb{E}_{p(y|x)}[T(x,y)] - \log \mathbb{E}_{p(y)}[e^{T(x,y)}]\right]. \tag{9}$$

Plugging into Eq. equation 5 and swapping the order of the integrals gives:

$$I(X;Y) \geq \sup_{T}\left[\mathbb{E}_{p(x,y)}[T(x,y)] - \mathbb{E}_{p(x)}\left[\log \mathbb{E}_{p(y)}\left[e^{T(x,y)}\right]\right]\right], \tag{10}$$

which is the first step towards the InfoNCE estimator (van den Oord et al., 2018).

In practice, the expectation over $p(y)$ is approximated using contrastive sampling. For a batch $\{(x_i, y_i)\}_{i=1}^{N}$:
- Treat $y_i$ as a *positive* sample from $p(y|x_i)$,
- Treat $\{y_j\}_{j \neq i}$ as *negative* samples from $p(y)$.

Using a Monte Carlo approximation of the expectations, one obtains:

$$I(X;Y) \geq I_{\mathrm{InfoNCE}}(X;Y) := \frac{1}{N}\sum_{i=1}^{N}\log \frac{e^{T(x_i,y_i)}}{\frac{1}{N}\sum_{j=1}^{N}e^{T(x_i,y_j)}}. \tag{11}$$

### A.1.2 BILINEAR CRITICS, GAUSSIAN VARIABLES, AND $I_{\text{CCA}}$

It is well known that, for jointly Gaussian variables, MI between two random variables can be written in terms of their nonzero canonical correlations, subject to keeping enough canonical pairs (Kullback, 1959; Gelfand & Iaglom, 1959; Huffmann & Mittelbach, 2022):

$$I_{\text{CCA}} = -\frac{1}{2} \sum_i^{K_Z} \log(1 - \rho_i^2), \tag{12}$$

where $\rho_i$ are the canonical correlations. Here, we show that this CCA estimate also emerges naturally as a special case of the *DV bound* on MI, with the concatenated bilinear (quadratic) critic family. This connects CCA to the same variational estimator framework as neural methods like InfoNCE and MINE, but with a different critic class.

Let $X \in \mathbb{R}^{K_X}$ and $Y \in \mathbb{R}^{K_Y}$ be jointly Gaussian with:

$$\begin{bmatrix} x \\ y \end{bmatrix} \sim \mathcal{N}\left(0, \begin{bmatrix} \Sigma_{XX} & \Sigma_{XY} \\ \Sigma_{YX} & \Sigma_{YY} \end{bmatrix}\right) := \mathcal{N}(0, \Sigma). \tag{13}$$

Using the same DV representation as for the MINE estimator, Eq. (7), we write:

$$I(X;Y) \geq \max_T \left\{ \mathbb{E}_{p(x,y)}[T(x,y)] - \log \mathbb{E}_{p(x)p(y)}[e^{T(x,y)}] \right\}. \tag{14}$$

As always, the globally optimal critic saturating the bound is $T^*(x,y) = \log \frac{p(x,y)}{p(x)p(y)} + c$, where $c$ is an arbitrary constant. For Gaussian data, this expression becomes:

$$T^*(x,y) = \frac{1}{2}\left[ x^\top \Sigma_{XX}^{-1} x + y^\top \Sigma_{YY}^{-1} y - \begin{pmatrix} x \\ y \end{pmatrix}^\top \Sigma^{-1} \begin{pmatrix} x \\ y \end{pmatrix} \right] + c. \tag{15}$$

We now define the canonical pairs. First, the whitened cross-covariance is

$$\mathcal{K} = \Sigma_{XX}^{-1/2} \Sigma_{XY} \Sigma_{YY}^{-1/2} = U\Lambda V^\top, \qquad \Lambda = \text{diag}(\rho_1, \ldots, \rho_{K_Z}), \tag{16}$$

where $U$ and $V$ are the matrices of left and right singular vectors of $\mathcal{K}$, and $\Lambda$ has canonical correlations on the diagonal. Then the canonical coordinates are

$$u = U^\top \Sigma_{XX}^{-1/2} x, \quad v = V^\top \Sigma_{yy}^{-1/2} y. \tag{17}$$

In these coordinates, the optimal critic is just the sum over independent canonical pairs:

$$T^\star(x,y) = \sum_{i=1}^{K_Z} \frac{\rho_i}{1 - \rho_i^2}\left[ u_i v_i - \frac{\rho_i}{2}\left(u_i^2 + v_i^2\right) \right] \tag{18}$$

(the expression under the sign of the sum can be verified by direct calculation for a bivariate normal distribution over $u_i$, $v_i$). Plugging Eq. (18) into Eq. (14) then gives MI in the form Eq. (12).

In other words, the DV optimal critic for the Gaussian distribution is a quadratic form, Eq. (18). Such a form belongs to a class of bilinear concatenated critics in Eq. (2), where $f$ concatenates its arguments and forms a bilinear expression from them, and $g$ and $h$ are linear operators, or simply identities.

We can also achieve the same result by directly calculating the optimal critic within a family of concatenated quadratic forms

$$T = z^\top W z, \quad z^\top = (x^\top, y^\top), \tag{19}$$

with the matrix $W$ such that both terms in the r.h.s. of Eq. (14) are finite. Then

$$\mathbb{E}_{p(x,y)}[z^\top W z] = \text{tr}(\Sigma W), \tag{20}$$

$$\mathbb{E}_{p(x)p(y)} e^{z^\top W z} = \frac{1}{\sqrt{\det(I - 2\Sigma_{\text{prod}} W)}}, \quad \text{where } \Sigma_{\text{prod}} = \begin{bmatrix} \Sigma_{XX} & 0 \\ 0 & \Sigma_{YY} \end{bmatrix}, \tag{21}$$

provided $I - 2\Sigma_{\text{prod}}W$ is positive definite. Now differentiating the DV bound, Eq. (14) w.r.t. $W$, we find the condition for $W^\star$, which optimizes the critic:

$$\Sigma - \left(\Sigma_{\text{prod}}^{-1} - 2W^\star\right)^{-1} = 0. \tag{22}$$

This results in

$$W^\star = \frac{1}{2}\left(\Sigma_{\text{prod}}^{-1} - \Sigma^{-1}\right), \tag{23}$$

which, sandwiched between $z^\top$ and $z$, again gives Eqs. (15) and (18).

Overall, these results say that the CCA estimate of MI emerges naturally from the bilinear (quadratic) concatenated critic family for Gaussian data within the DV framework.

## A.2 PROBABILISTIC CRITICS: VARIATIONAL SYMMETRIC INFORMATION BOTTLENECK (VSIB)

The Variational Symmetric Information Bottleneck (VSIB)[2] can formalize MI estimation as a form of probabilistic dimensionality reduction. It introduces a latent representation for each variable—$Z_X$ and $Z_Y$—produced by separate stochastic encoders from $X$ and $Y$, respectively. Mutual information is then estimated between these latent representations using *any* neural MI estimator of choice.

This leads to the following objective (Abdelaleem et al., 2025):

$$L_{\text{EST}_{\text{VSIB}}} = I^E(X; Z_X) + I^E(Y; Z_Y) - \beta I_{\text{EST}}^D(Z_X; Z_Y), \tag{24}$$

where $I^E(\cdot\,;\cdot)$ are encoder regularization terms, and $I_{\text{EST}}^D(Z_X; Z_Y)$ is the mutual information in the latent space estimated using a particular chosen neural estimator (e.g., InfoNCE or SMILE).

Each encoder term is computed as:

$$I^E(X; Z_X) \approx \frac{1}{N}\sum_{i=1}^{N} D_{\text{KL}}(p(z_x|x_i)\|r(z_x))$$

$$\approx \frac{1}{2N}\sum_{i=1}^{N}\left[\text{Tr}(\Sigma_{Z_X}(x_i)) + \|\vec{\mu}_{Z_X}(x_i)\|^2 - k_{Z_X} - \ln\det(\Sigma_{Z_X}(x_i))\right], \tag{25}$$

where $\vec{\mu}_{Z_X}(x)$ and $\Sigma_{Z_X}(x)$ parameterize the mean and covariance of the encoder distribution $p(z_x|x)$, and $k_{Z_X}$ is the latent dimensionality. The same form is used for $I^E(Y; Z_Y)$. The scalar $\beta > 0$ controls the trade-off between the regularization terms and the estimated information. In our experiments, we used $\beta = 512$, which strongly prioritizes $I_{\text{EST}}^D$ while still regularizing the encoder mappings. Other large values of $\beta$ yielded similar results (not shown).

In the limit where the encoders $p(z_x|x)$ and $p(z_y|y)$ collapse to delta distributions, the $I^E$ terms converge to the entropy $H(z_x)$ and $H(z_y)$, respectively. These terms diverge to infinity and do not change as a function of the embedding and thus do not affect the estimation of information between the latent variables, so that Eq. (24) recovers the standard deterministic neural MI estimators.

While VSIB naturally aligns with separable critics (i.e., $T(x,y) = \langle g(x), h(y)\rangle$), the framework can be generalized to concatenated critics. Specifically, one can define a latent variable $Z$ such that:

$$Z \sim \mathcal{N}(\mu([x,y]), \Sigma([x,y])), \tag{26}$$

with neural networks $\mu(\cdot)$ and $\Sigma(\cdot)$ operating on the joint input $[x,y]$. The loss becomes:

$$L = I^E([X,Y]; Z) - \beta I_{\text{EST}}^D(Z(X,Y)), \tag{27}$$

where $I_{\text{EST}}^D(Z(X,Y))$ is just another way to write $I(X;Y)$ for a concatenated critic. While this version is less directly interpretable in terms of a variable compression, it is straightforward to implement and was used in Figs. 1 and 2 as the probabilistic variant of SMILE and InfoNCE (denoted with $\text{EST}_{\text{VSIB}}$).

---

[2]This is an instance of the more general, Deep Multivariate Information Bottleneck Framework (Friedman et al., 2013; Abdelaleem et al., 2025). In this framework, one specifies a compression/encoder graph that is traded off against a generative/decoder graph. Each graph is then transformed into an information bound that can be optimized.

## A.3 THE EARLY STOPPING HEURISTIC

To define the stopping heuristic, note both the training and the test data are sampled from the same $p(x, y)$. In practice, expectations for every estimator EST, such as in Eq. (7, 8, 11), are implemented with empirical sampling. That is, we form empirical densities $\pi_{\text{train}} = \frac{1}{N_{\text{train}}} \sum_{i=1}^{N_{\text{train}}} \delta(x - x_i, y - y_i)$, and similar for $\pi_{\text{test}}$ for the test data. Then, for example, the MINE estimator is implemented as,

$$L_{\text{EST}}(\pi_{\text{train}}, T) = \mathbb{E}_{\pi_{\text{train}}(x,y)}[T(x,y)] - \log \mathbb{E}_{\pi_{\text{train}}(x)\pi_{\text{train}}(y)}\left[e^{T(x,y)}\right], \tag{28}$$

and similarly for the other estimators. Then the "train" and "test" MI values are defined via:

$$T_{\text{train}}^* = \arg\max_T L_{\text{EST}}(\pi_{\text{train}}, T), \tag{29}$$

$$I_{\text{EST, train}} = L_{\text{EST}}(\pi_{\text{train}}, T_{\text{train}}^*), \tag{30}$$

$$I_{\text{EST, test—train}} = L_{\text{EST}}(\pi_{\text{test}}, T_{\text{train}}^*). \tag{31}$$

For completeness, we also define the true (typically unknown) mutual information as $I_{\text{true}}$, with the globally optimal critic

$$T^*(x, y) = \log \frac{p(x,y)}{p(x)p(y)} + c, \tag{32}$$

where $c$ is an arbitrary constant.

With these definitions, we use the following procedure:
1. Train the estimator on $\pi_{\text{train}}$ for several epochs in each step of the algorithm, see Appx. A.6.
2. After each such step, freeze $T_{\text{train}}^*$, and evaluate both $I_{\text{EST, train}}$ and $I_{\text{EST, test—train}}$ for whichever estimator EST is being used.
3. Select $\hat{T} = T_{\text{train}}^*$ corresponding to the cycle where $I_{\text{EST, test—train}}$ is maximal.
4. Report $I_{\text{EST, train}}$ evaluated at $\hat{T}$ as the final MI estimate.

This procedure regularizes overfitting. While one could attempt architectural regularization (e.g., dropout or weight decay) to stabilize training, such strategies offer no clear way to assess the trustworthiness of the resulting estimate. In contrast, the max-test heuristic explicitly favors models that perform best on unseen data.

Here we justify this heuristic, which may appear to conflict standard machine learning practice, which typically report test results. However, we argue that in the case of neural MI estimators, our approach is correct. Fundamentally, this is because MI is not a linear functional of the underlying data distribution, unlike many other expectation values, which are linear in the measure. Therefore, unbiased estimates of the distribution do not yield unbiased MI estimates (Nemenman et al., 2001). Because of this, it is well known, for example, that resampling approaches, such as bootstrap and cross-validation, result in errors in biases in MI estimation (Holmes & Nemenman, 2019). Here we argue that similar results extend to the current case. The key insight is that MI estimation is an *estimation of a functional* not a *prediction* task. An estimator is trained to estimate this functional with small bias on the *training* set, and its performance on the test set does not necessarily approximate the functional. We argue this in a few ways.

**Biases of estimators.** General bounds showing that $I_{\text{EST,test|train}}$ typically understimates $I_{\text{true}}$ and hence should not be used as a reported estimate cannot exist without additional strong assumptions about $T$. To see this, note a counter-example: if the test set consists of just one sample, and the training set has many, and $T$ is optimized over the class that contains just a single peak, but at different locations in $(x, y)$, then the test MI can be very large (when the critic peak matches the single sample), while the training MI will be low, and either can over- or under-estimate $I_{\text{true}}$. Thus to argue for the max-test heuristic, we need additional assumptions.

There are many variants of similar such assumptions, all starting with assuming that $T_{\text{train}}^* \approx T^*$ (that is, the trained optimal critic is almost globally optimal), and all resulting in $I_{\text{EST,test|train}} \leq I_{\text{true}}$. We do not know which of the assumptions would be convincing to the reader, and so here we give just one loose proof arguing that the test value is an underestimate and should not be reported. Since test and training sets are taken from the same distribution, and assuming they are the same size

$$\mathbb{E}_{p(x,y)} I_{\text{EST,test|train}} \equiv \mathbb{E}_{p(x,y)} L_{\text{EST}}(\pi_{\text{test}}, T_{\text{train}}^*)$$
$$\leq \mathbb{E}_{p(x,y)} L_{\text{EST}}(\pi_{\text{test}}, T_{\text{test}}^*) = \mathbb{E}_{p(x,y)} L_{\text{EST}}(\pi_{\text{train}}, T_{\text{train}}^*) \equiv \mathbb{E}_{p(x,y)} I_{\text{EST,train}}, \tag{33}$$

where the inequality is due to the maximization in the definition of $I_{\text{EST}}$. In other words, the test MI is expected to be lower than the training value for statistically similar test and training sets. In particular, this means that if $T^*_{\text{train}} = T^*$ and $I_{\text{EST,train}} = I_{\text{true}}$, then $\mathbb{E}_{p(x,y)} I_{\text{EST,test}|\text{train}} \leq I(X,Y)$. In other words, if the set of critics that the neural network optimizes over includes the critic that saturates the DV bound, and sampling and the training algorithm are such that the globally optimal critic is found during the optimization, then the test value of the estimator will be biased down, on average.

**Other approaches.**    The DEMINE and meta-DEMINE MI estimators (Lin et al., 2019) also attempt to produce more efficient estimators by splitting data into validation and training data sets. The training data sets can be much larger and even use task augmentation to allow the critic to be well-learned. The data efficiency of estimating information on the validation set is decoupled from learning the critic and thus can lead to better efficiency for estimating information on the validation set. The problem with this procedure is the assumption that the training dataset is large enough for the critic to be well learned and provides no way to determine when the critic is well learned.

**Empirical observations.**    Empirically, in all cases we reported in the paper and many others we tried but not reported, we observe that the test MI estimate is consistently biased downward (cf. Fig. 3). In contrast, the train MI estimate, evaluated using the best-performing model on the test set (i.e., the model checkpoint that achieved the highest test MI before overfitting), provides a more accurate estimate of the true MI in controlled synthetic setups (e.g., cf. Fig. 4).

### A.4    DETAILED GUIDELINES FOR ESTIMATING MI

Here we provide detailed description of the workflow for MI estimation, outlined in the main text.

**1. Choosing an Estimator.**    Select the estimator (EST) appropriate for the expected range of MI and data complexity. For small expected information content (e.g., well below $\log(\text{batch size})$) with enough samples to distinguish distinct information levels, InfoNCE is a good choice. However, InfoNCE saturates as information increases unless the batch size is scaled appropriately. In such high-MI regimes, SMILE becomes preferable. However, we observe that when information is high, SMILE significantly overfits. Then using probabilistic embeddings regularized via VSIB improves performance (see Fig. 4). We thus recommend SMILE–VSIB when high information content and overfitting are both concerns.

**2. Network Design and Critic Architecture.**    Critic architecture should match the data modality. We use MLPs for simplicity and consistency across experiments, but CNNs or transformers (Vaswani et al., 2017) may be more appropriate in other tasks. In practice, any architecture that effectively captures the statistics of the input may be used, as the critic is simply an embedding-to-MI pipeline. We have verified that CNNs also work in our setup on Noisy MNIST image data from Sec. 4.3 (not shown).

Separable critics offer simplicity, modularity, and support for modality-specific parametrizations (e.g., using CNN for images and transformers for text), as well as faster computation via dot products. They also support variable embedding dimensionality, which plays a crucial role in practice. In contrast, concatenated critics jointly embed $(X,Y)$, capturing more complex interactions, but at the cost of more computation. They also do not provide a way to vary the latent dimensionality $k_Z$ explicitly. We recommend using concatenated critics only if the sample size is large, and computational resources are not an issue. Additionally, they should only be used if the critic dimensionality is not interesting to know, and the data modalities are homogeneous; so as to avoid mixing units. Finally, note that, in linear regimes (see Figs. 1 and 2), linear critics suffice and are significantly more efficient.

**3. Subsampling and Max-Test Stopping Heuristic.**    Start with the separable critic embedding dimension of $k_Z = 1$. Subsample the dataset into $\gamma$ non-overlapping subsets and evaluate the estimator on each subset. Start with $\gamma = 1$ (i.e., the full dataset) and compute the MI using the max-test stopping heuristic (Fig. 3). We typically train up to 100 epochs with an additional early stopping rule if test performance stops improving for 50 epochs. More generally, the criteria should be that the training is long enough to notice saturation or decline of the test curve. Then, for $\gamma = 2$, split the dataset into two halves and compute MI on each. Continue increasing $\gamma$ up to 10 (one tenth

of the full dataset in each subset). Since datasets often cannot be evenly divided, allocate $\lfloor N/\gamma \rfloor$ samples to each of the first $\gamma - 1$ subsets and assign the remainder to the final subset.

**4. Mean and Variance Estimation.** For each $\gamma$, compute the sample mean $\bar{I}_{\text{EST}}^{k_Z}(\gamma)$ and standard deviation $\sigma^{k_Z}(\gamma)$ of the resulting MI values across subsets.

**5. Embedding Dimensionality Search.** Repeat steps 3 and 4 for increasing values of embedding dimensionality $k_Z$ (in separable critics). As $k_Z$ increases, MI estimates should rise until they plateau. Identify the minimal $k_Z$ beyond which estimates no longer increase significantly—this defines $k_Z^*$. Slight overestimation of $k_Z^*$ is acceptable. Note that overly large $k_Z$ may lead to severe undersampling and collapse of the information values; avoid being in this regime. This approach is flexible; other criteria to determine $k_Z^*$ may be more suitable in specific contexts. For concatenated critics, skip this step as $k_Z$ is fixed.

**6. Weighted Fitting and Linear Extrapolation to $\gamma \to 0$.** After selecting the embedding dimensionality $k_Z^*$, we estimate the MI by extrapolating the estimates taken at different data fractions to the infinite data regime, which, for a well-trained, sufficiently expressive critic, should correspond to the true MI. We do this by fitting the curve $\bar{I}_{\text{EST}}^{k_Z}(\gamma)$ versus $\gamma$. Because we have unequal numbers of samples at each $\gamma$ (i.e., different number of subsets), a standard ordinary least squares (OLS) regression would inappropriately put too much weight to large $\gamma$. Instead, we use a weighted least squares (WLS) approach.

Traditional best practice (Holmes & Nemenman, 2019) suggests using inverse variance across all subsets at a given data fraction as weights in WLS, with the variance smoothed via an OLS model fit to the subset variances across all fractions. This approach assumes that the estimator variance scales as $1/N$ (i.e., $\propto \gamma$), as seen with traditional estimators like $I_{\text{KSG}}$. However, this does not hold for neural MI estimators. We have verified that these estimators exhibit relatively uniform variance across $\gamma$ (not shown), which is likely dominated not by sample size but by stochasticity in optimization, random initializations, and inherent signal structure (Song & Ermon, 2019). Thus, we instead suggest assigning weights to each subset proportional to its $1/\gamma$. For $\gamma = 1$, the single dataset is then given full weight; for $\gamma = 2$, each of the two halves is weighted by $1/2$, and so on.

We then suggest fitting a WLS quadratic model:

$$\bar{I}_{\text{EST}}^{k_Z}(\gamma) = a_2 \gamma^2 + a_1 \gamma + a_0. \tag{34}$$

To determine whether a linear approximation is valid, we suggest computing $\delta = |a_2/a_1|$, the relative strength of the quadratic term compared to the linear one at $\gamma = 1$ (full data). If $\delta > 0.1$, the tail of the curve (i.e., large $\gamma$) exerts too much nonlinear influence. Thus one should iteratively prune the largest $\gamma$ value and the corresponding data from the fit. One then should refit the quadratic model, recompute $\delta$, and continue removing the large $\gamma$ values until either $\delta < 0.1$, until a smaller range $\tilde{\gamma}$ is found, over which the information curve is approximately linear.

If in this process the range $\gamma \leq 5$ is reached, then the information cannot be reliably extrapolated linearly. Nonlinear fits should not be used: if a term quadratic in $\gamma$ (equivalently, quadratic in $1/N$) is similar in magnitude to the linear term, then all the higher order terms are likely large too. There is no reason to truncate the Taylor series at low orders, and hence extrapolation should not be performed. The estimator should return "estimation unreliable" and quit.

If, on the contrary, the reduced range $\tilde{\gamma}$ remains large (the range did not need to be reduced for any NN estimator in any of the figures in this paper), reliable estimation is possible. We suggest then to perform a final linear WLS fit over $\tilde{\gamma}$:

$$\bar{I}_{\text{EST}}^{k_Z}(\tilde{\gamma}) \approx b\tilde{\gamma} + c. \tag{35}$$

**7. MI Estimation and Prediction Interval.** Use the intercept of the linear WLS fit of $\bar{I}_{\text{EST}}^{k_Z^*}(\tilde{\gamma})$ vs $\tilde{\gamma}$ dependence to estimate MI in the $\tilde{\gamma} = 0$ limit: $\hat{I} = c$. This provides an estimate at the infinite data limit and follows a procedure similar to Strong et al. (1998). Alternatively, one can interpolate to $\gamma = 1$ to find an MI for the full dataset size, less susceptible to statistical fluctuations. The corresponding prediction interval $\Delta I$ from the linear fit quantifies uncertainty in either estimate.

Note that this reported value may still be an underestimate of MI: if information is contained in features at small scales, inaccessible at the full sample size, no general purpose MI estimation algorithm will be able to recover it.

## A.5 ESTIMATING SAMPLE SIZE REQUIRED FOR MUTUAL INFORMATION ESTIMATION IN THE LATENT VARIABLE MODEL

To make sense of the sample size $N$ that allows MI estimation in Fig. 5, it is helpful to recall the spiked covariance model from random matrix theory (Potters & Bouchaud, 2020). There, the data covariance matrix has the structure $\Sigma = \sigma_n^2(\mathbb{I} + \theta v^\top v)$, where $\sigma_n^2$ is the per-coordinate noise variance, and $v$ is a low-rank matrix with orthonormal columns, corresponding to independent signals on the background of Gaussian noise. Then the spike signal of magnitude $\theta$ separates from the sampling-induced spurious background correlations when $\theta > \sqrt{k/N}$, where $k$ is the observed dimensionality (Baik et al., 2005). The overlap of the largest eigenvector of the data covariance matrix with the spike vectors $v$ grows continuously beyond this threshold.

The latent model we use in this work can be cast in this language by considering $2K_Z$ dimensional Gaussian random variables $(x^\top, y^\top)$, where each pair $x_i, y_i$ is produced from a latent variable $z_i$, and otherwise $\mathbb{E}x_i x_j = \mathbb{E}x_i y_j = \mathbb{E}y_i y_j = 0$ if $i \neq j$. We have $v^\top = (1/\sqrt{2}, 1/\sqrt{2})$. The covariance matrix of the $i$th pair is

$$\Sigma_i = \begin{pmatrix} 1 + \theta_i/2 & \theta_i/2 \\ \theta_i/2 & 1 + \theta_i/2 \end{pmatrix} = \begin{pmatrix} 1 + \theta_i' & \theta_i' \\ \theta_i' & 1 + \theta_i' \end{pmatrix},, \tag{36}$$

with $\theta' = \theta/2$, and the correlation coefficient $\rho_i = \theta_i'/(1 + \theta_i')$. Since, for each pair, $I_i = -1/2 \log(1 - \rho_i^2)$,

$$\theta_i' = \frac{1}{1 - (1 - 2^{-2I_i})^{1/2}} - 1. \tag{37}$$

Thus, if $K_Z \gg 1$, and $\operatorname{rank} v \ll 2K_Z$ (neither of these conditions is strictly true in our model), then the detection limit for each signal component is

$$N > N_Z^* = \frac{2K_Z}{\theta^2} = \frac{2K_Z}{(2\theta')^2} = \frac{K_Z \left(1 - \sqrt{1 - 2^{-2I_i}}\right)^2}{2\left(1 - 2^{-2I_i}\right)} = \frac{K_Z \left(1 - \sqrt{1 - 2^{-2I/K_Z}}\right)^2}{2\left(1 - 2^{-2I/K_Z}\right)}, \tag{38}$$

where the last step is because total MI is evenly divided over $K_Z$ pairs. In the weak signal limit, $2I/K_Z \ll 1$, this becomes

$$N \gtrsim \frac{K_Z^2}{I \ln 2} \left(1 - 2\sqrt{2 \ln 2 I/K_Z}\right), \tag{39}$$

which scales as $\sim K_Z^2$. This is because higher dimensionality makes signal detection harder, and it also lowers information per dimension. Intuitively, reliable MI estimation cannot begin until each spike is first detected, so we expect $N > N_Z^*$ before the estimated MI becomes significantly nonzero. Further, $N$ must grow well beyond $N_Z^*$ for the detected eigenvalues of the covariance matrix within this linear model to become close to the spike signals, thus allowing MI to reach its true value.

The bound $N_Z^*$ is optimistic: it ignores interactions among multiple spikes and effects of the nonlinear embedding into the full $2K$-dimensional data space, thus assuming that no samples are spent learning that embedding. Repeating the spike-detection calculation with the full dimensionality $K$ yields a looser condition $N > N^*$, appropriate for detecting a linear spike in the data space. One would therefore expect the practical requirement to satisfy $N \gg N^* > N_Z^*$, since the critic must resolve the signal within a $2K$-dimensional nonlinear map. Figure 5 shows the opposite: accurate MI becomes possible soon after as $N$ exceeds the tighter latent bound $N_Z^*$, but before the $N^*$ bound is reached, at least for moderate $K_Z$.

## A.6 IMPLEMENTATION DETAILS

**Critic Architectures.** To simplify comparisons, for all synthetic experiments (i.e., all figures except Fig. 7), we use feedforward multi-layer perceptrons (MLPs) with two hidden layers of 256 units each, initialized using Xavier uniform initialization (Glorot & Bengio, 2010), and using leaky ReLU

activations. For the MNIST dataset, due to the added difficulty of the task, we use deeper networks with four hidden layers of width 512. We also used CNNs for this dataset with similar results (not shown).

Recall that our general critic has the structure $T(x, y) = f(g(x), h(y))$. Separable critics use one MLP for $g(x)$ and one for $h(y)$; the dot product of their outputs defines $T(x, y)$. For concatenated critics, $g$ and $h$ are identities, and $f$ is realized as an MLP with the input $[x, y]$ (dimension $K_X + K_Y$) and output dimension 1. In both cases, the hidden width is set to the maximum of the widths used for $x$ and $y$, which are equal in our experiments.

For probabilistic critics, the architecture is identical to the deterministic case, except that the final layer is split into two heads that parameterize the mean and log-variance of a Gaussian conditional distribution of the embedding. Sampling is done via the standard reparameterization trick.

The choice of MLPs to implement the critics is just for convenience, and is not a crucial aspect of our approach. Different architectures can be used, as long as they produce a scalar critic. For example, convolutional NNs or transformers are likely to be better choice for the neural critics instead of MLPs for image or text data, respectively (see Appx. A.4).

**Training Details.** All models are trained using the ADAM optimizer with a learning rate of $5 \times 10^{-4}$ and a batch size of 128, or the full dataset if it is smaller in size than 128. Each model is trained for a maximum of 100 epochs. Early stopping is applied if the test MI estimate, always evaluated on a fixed heldout batch of size 128 (even when the training set contains fewer than 128 samples), does not improve for 50 consecutive epochs or if the maximum number of epochs is reached.

**Other Hyperparameters.** All unspecified hyperparameters use their default values as implemented in the *PyTorch* v2.0.1, *SciPy* v1.11.1, *cca_zoo* (Chapman & Wang, 2021) v2.3.11, and *statsmodels* v0.14.2 libraries.

**Compute Resources** All experiments were conducted on AWS instances. We used CPUs for $I_{\text{CCA}}$ and $I_{\text{KSG}}$, and GPUs for all neural network-based estimators. The primary instance types included `h200-8-gm1128-c192-m2048`, `a100-8-gm320-c96-m1152`, and `l40s-8-gm384-c192-m1536`.

As a reference, training a single neural estimator for one embedding dimensionality $k_Z$ on the MNIST dataset used in Fig. 7 (the $\gamma = 1$ point with $2^{14} \approx 16\text{k}$ samples) takes roughly 20 seconds. Computing across all tested $k_Z$ values takes approximately 100 seconds. Subsampling experiments take a comparable amount of time; for example, evaluating two half-sized datasets (2 subsets at $\gamma = 2$) takes approximately the same time as training on the full dataset. Since this procedure (as illustrated in Fig. 7) represents the main recommended pipeline for mutual information estimation, we report its runtime in detail.

For completeness, other figures—such as the left panel of Fig. 5—required up to 1,000 seconds for the highest sample count ($\approx 65\text{k}$), 10 trials, and multiple $k_Z$ values. The previous data point with half the number of samples ($\approx 32\text{k}$) took approximately 500 seconds.

For $I_{\text{CCA}}$ in Fig. 2, computing MI at a single MI level and embedding dimension $k_Z$ took approximately 70 seconds. In Appx. A.7.2, we benchmark our approach against other methods; so here we report their compute requirements as well. For $I_{\text{KSG}}$ (as used in Appx. Fig. 9), the first panel took approximately 50 seconds for the entire sweep; the second panel, which had more samples, took around 500 seconds. Both were run on CPU nodes. $I_{\text{LMI}}$ took about 5 minutes to run on a desktop CPU. $I_{\text{Geodesic}}$ took 2.25 days to run on the CPU nodes of the cluster and needed more than 128 GB of RAM to complete. Neither of these estimates sample-size dependent biases or confidence intervals. That is, our approach compares favorably to others in terms of computing needs.

Overall, the total compute time, including exploratory and failed runs that led to this work, is estimated on the order of 500 compute hours across CPUs and GPUs.

**Smoothing and the Stopping Heuristic.** To avoid stopping based on high-frequency fluctuations during training, we first smooth the training and test curves using the median filter to remove outliers. We use the filter window size of 40 steps in Figs. 1 and 2, where training is done on very large number of batches/steps, and a smaller window of 5 in all other Figures. With outliers removed, we further

smooth the results with a Gaussian filter with the standard deviation $\sigma = 1$. This yields the smooth test and training curves shown throughout the paper. We note that this smoothing strategy is heuristic, and other methods may be more appropriate in different settings.

**Final result reporting.** While it is possible to evaluate $I_{\text{EST, train}}$ over the full training dataset, we found that evaluating it on a representative batch is sufficient for early stopping and reporting. Larger-scale averaging can be performed at evaluation time if memory allows. However, for completeness, we performed such an evaluation over the full training dataset for the task mentioned later in Sec. 2 and the performance is consistent, albeit with technical caveats to prevent memory related crashes.

### A.7 EVALUATIONS WITH OTHER ESTIMATORS AND DATASETS

In this section, we provide additional evaluations of our pipeline beyond the experiments reported in the main text. Specifically, we study: (i) alternative estimators in the infinite-data, low-dimensional regime, which motivated our choice of InfoNCE and SMILE; (ii) other popular estimators, including KSG (Kraskov et al., 2004), Sliced MI (SMI) (Goldfeld & Greenewald, 2021), Geodesic MI (Marx & Fischer, 2022), and LMI (Gowri et al., 2024), in the finite-data regime of our teacher network task (Fig. 4); and (iii) evaluations on the benchmark datasets introduced in Czyz et al. (2023), which allows us to compare the traditional use of InfoNCE with its modification in our pipeline.

#### A.7.1 INFINITE DATA REGIME

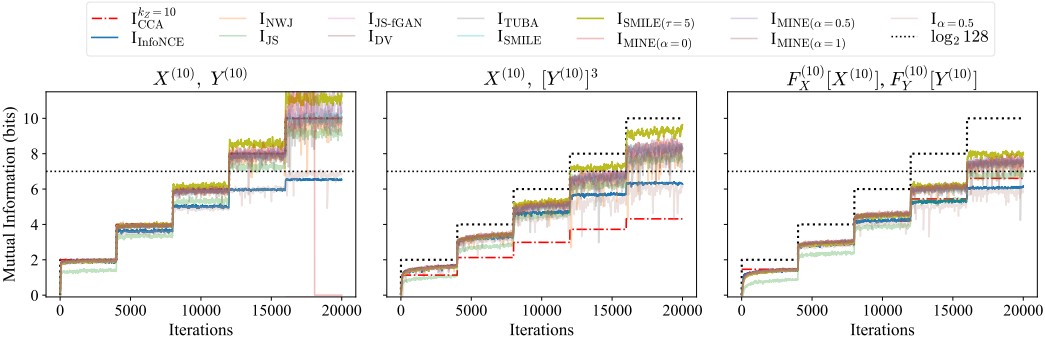

Figure 8: **Comparison of MI estimators in the low-dimensional, infinite-data regime.** We replicate the setup of Fig. 1 but include a broader selection of estimators (implementations adapted from Song & Ermon (2019); Poole et al. (2019)). True MI is varied in discrete steps, and all estimators are run with identical batch size and sample schedules. SMILE tracks the ground truth most closely but with higher variance, while InfoNCE is biased downward at high MI yet remains stable. Other estimators either diverge during training or fail to capture the correct trends. This motivates focusing on SMILE and InfoNCE in our main study.

#### A.7.2 FINITE DATA REGIME

**KSG.** The KSG estimator (Kraskov et al., 2004) with code adapted from Czyz et al. (2023), was used to estimate information contained in $K = 500$-dimensional data generated by teacher networks with latent dimensionality $K_Z = 10$ and true MI of 4 bits, as used in Fig. 6 for neural estimators. Best practices, were used to estimate the MI with KSG estimator (Holmes & Nemenman, 2019) at different numbers of nearest neighbors $k$. Specifically, as for NN estimators, data was partitioned into $\gamma$ non-overlapping partitions and estimates for each partition were found. The mean information and standard deviation using $\gamma$ partitions were plotted versus the number of partitions at different values of $k$ in Fig. 9. The KSG estimator does not asymptotically approach the correct value of information at $\gamma = 0$, corresponding to the infinite data limit, and linear extrapolations (done the same way as in Fig. 6) is unreliable for KSG in high dimensions. Best practice is to not use the estimator in this case (Holmes & Nemenman, 2019).

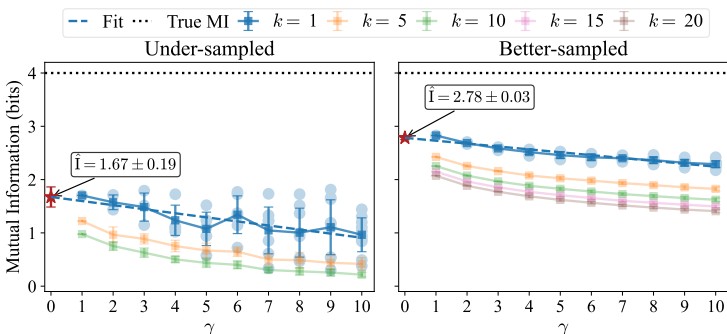

Figure 9: **Failure of the KSG estimator in high dimensions.** Estimation of $I(X; Z)$ with KSG (Kraskov et al., 2004) on $K = 500$-dimensional teacher data with latent dimension $K_Z = 10$ and ground-truth MI of 4 bits. Left: undersampled ($N = 256$) regime; Right: well-sampled ($N = 2^{14}$). For different nearest-neighbor values $k$, MI estimates remain far below ground truth (1–2.8 bits) and do not converge under extrapolation. In contrast, neural estimators recover near-perfect values (Fig. 6). This illustrates that KSG is unreliable for high-dimensional data, consistent with Holmes & Nemenman (2019).

**Sliced MI.**   Sliced Mutual Information (SMI) (Goldfeld & Greenewald, 2021) does not estimate true MI but a surrogate quantity that violates the data processing inequality. While SMI may be useful for representation learning, comparing it directly to MI estimators is misleading. On our high-dimensional teacher network benchmark, available implementations yielded values far below ground truth (e.g., 1.13 bits vs. 4 bits), highlighting its inapplicability as a general MI estimator.

**Geodesic MI.**   Geodesic-MI (Marx & Fischer, 2022) proved computationally prohibitive: runs required multiple days and $> 128$GB RAM. For the better-sampled case ($N = 2^{14}$), it produced estimates of 1.86 and 1.72 bits (for $k = 3, 10$), far below the 4-bit ground truth, showing both poor scalability and low accuracy compared to our approach.

**LMI.**   Latent MI (LMI) estimators (Gowri et al., 2024) assume linear-Gaussian structure and are unreliable in nonlinear settings. On our teacher network task (true MI 4 bits), LMI gave 0.65 bits (undersampled) and 1.6 bits (better-sampled), versus our $4.1 \pm 0.2$ and $3.7 \pm 0.1$. Unlike LMI's two-step compression–estimation procedure, our end-to-end training captures nonlinear dependencies directly.

### A.7.3   OTHER BENCHMARK DATASETS

Czyz et al. (2023) introduced a diverse suite of datasets ($1 \leq K \leq 50$) with analytically tractable MI values. They reported that InfoNCE generally outperformed alternatives (Figure 2 in their paper). We revisit these datasets (10,000 samples) using the same critic architecture as in their study (a concatenated critic with two hidden layers of width 16), but apply our bias correction, error bar estimation pipeline with held-out testing (9k train, 1k test samples).

Table 2 compares ground-truth MI, results from Czyz et al. (2023), and our pipeline. Grey rows mark unreliable fits ($\delta > 0.1$ or $\gamma_{\max} \leq 5$). Our method consistently matches or exceeds baseline InfoNCE performance while uniquely providing confidence intervals.

For challenging datasets such as spirals, performance improves substantially with more expressive critics (hidden layers of width 256). Table 3 illustrates this case. Overall, our pipeline not only matches or surpasses state-of-the-art results, but is the only one offering error bars and internal consistency checks.

Table 2: **Benchmark results on datasets from Czyz et al. (2023).** True MI, results reported by Czyz et al. (2023) using simple InfoNCE implementation, and our pipeline with error bars. Grey rows indicate unreliable fits ($\delta > 0.1$ or $\gamma_{max} < 5$). Our method consistently achieves estimates similar to or better than (reported in **boldface**) the rivals, while uniquely reporting confidence intervals. Results reported in *nats* to match Czyz et al. (2023).

| Task | True MI | simple InfoNCE | Ours | $\gamma_{max}$ | $\delta$ |
|---|---|---|---|---|---|
| Uniform $1 \times 1$ (additive noise=0.1) | 1.71 | 1.7 | $1.67 \pm 0.03$ | 10 | 0.041 |
| Uniform $1 \times 1$ (additive noise=0.75) | 0.33 | 0.3 | $0.33 \pm 0.02$ | 10 | 0.026 |
| Bimodal $1 \times 1$ | 0.41 | 0.4 | $0.39 \pm 0.02$ | 5 | 0.195 |
| Bivariate normal $1 \times 1$ | 0.41 | 0.4 | $0.39 \pm 0.03$ | 9 | 0.087 |
| Asinh @ Student-t $1 \times 1$ (dof=1) | 0.22 | 0.2 | $0.24 \pm 0.02$ | 9 | 0.064 |
| Asinh @ Student-t $2 \times 2$ (dof=1) | 0.43 | 0.4 | $0.40 \pm 0.02$ | 9 | 0.068 |
| Asinh @ Student-t $3 \times 3$ (dof=2) | 0.29 | 0.2 | $\mathbf{0.26 \pm 0.03}$ | 10 | 0.098 |
| Asinh @ Student-t $5 \times 5$ (dof=2) | 0.45 | 0.3 | $0.34 \pm 0.07$ | 10 | 0.058 |
| Half-cube @ Bivariate normal $1 \times 1$ | 0.41 | 0.4 | $0.39 \pm 0.01$ | 10 | 0.077 |
| Half-cube @ Multinormal $25 \times 25$ (2-pair) | 1.02 | 0.8 | $\mathbf{1.02 \pm 0.07}$ | 5 | 0.234 |
| Half-cube @ Multinormal $3 \times 3$ (2-pair) | 1.02 | 1.0 | $1.00 \pm 0.02$ | 10 | 0.083 |
| Half-cube @ Multinormal $5 \times 5$ (2-pair) | 1.02 | 1.0 | $1.01 \pm 0.03$ | 9 | 0.062 |
| Multinormal $2 \times 2$ (dense) | 0.29 | 0.3 | $0.30 \pm 0.02$ | 10 | 0.086 |
| Multinormal $25 \times 25$ (dense) | 1.29 | 1.2 | $\mathbf{1.26 \pm 0.03}$ | 10 | 0.03 |
| Multinormal $3 \times 3$ (dense) | 0.41 | 0.4 | $0.40 \pm 0.02$ | 10 | 0.093 |
| Multinormal $5 \times 5$ (dense) | 0.59 | 0.6 | $0.60 \pm 0.03$ | 10 | 0.098 |
| Multinormal $50 \times 50$ (dense) | 1.62 | 1.4 | $\mathbf{1.60 \pm 0.03}$ | 10 | 0.062 |
| Multinormal $2 \times 2$ (2-pair) | 1.02 | 1.0 | $1.02 \pm 0.03$ | 10 | 0.059 |
| Multinormal $25 \times 25$ (2-pair) | 1.02 | 0.9 | $\mathbf{0.98 \pm 0.05}$ | 5 | 0.162 |
| Multinormal $3 \times 3$ (2-pair) | 1.02 | 1.0 | $1.00 \pm 0.02$ | 5 | 0.166 |
| Multinormal $5 \times 5$ (2-pair) | 1.02 | 1.0 | $1.02 \pm 0.02$ | 7 | 0.045 |
| Normal CDF @ Bivariate normal $1 \times 1$ | 0.41 | 0.4 | $0.37 \pm 0.02$ | 10 | 0.034 |
| Normal CDF @ Multinormal $25 \times 25$ (2-pair) | 1.02 | 0.8 | $\mathbf{0.94 \pm 0.12}$ | 9 | 0.009 |
| Normal CDF @ Multinormal $3 \times 3$ (2-pair) | 1.02 | 0.9 | $0.92 \pm 0.04$ | 10 | 0.048 |
| Normal CDF @ Multinormal $5 \times 5$ (2-pair) | 1.02 | 0.9 | $0.93 \pm 0.04$ | 10 | 0.046 |
| Spiral @ Multinormal $25 \times 25$ (2-pair) | 1.02 | 0.7 | $\mathbf{0.97 \pm 0.24}$ | 9 | 0.022 |
| Spiral @ Multinormal $3 \times 3$ (2-pair) | 1.02 | 0.6 | $0.69 \pm 0.04$ | 10 | 0.037 |
| Spiral @ Multinormal $5 \times 5$ (2-pair) | 1.02 | 0.6 | $0.64 \pm 0.04$ | 10 | 0.035 |
| Spiral @ Normal CDF @ Multinormal $25 \times 25$ (2-pair) | 1.02 | 0.8 | $0.89 \pm 0.10$ | 6 | 0.096 |
| Spiral @ Normal CDF @ Multinormal $3 \times 3$ (2-pair) | 1.02 | 0.9 | $0.89 \pm 0.03$ | 10 | 0.042 |
| Spiral @ Normal CDF @ Multinormal $5 \times 5$ (2-pair) | 1.02 | 0.9 | $0.88 \pm 0.04$ | 9 | 0.064 |
| Student-t $1 \times 1$ (dof=1) | 0.22 | 0.1 | $\mathbf{0.21 \pm 0.03}$ | 5 | 0.195 |
| Student-t $2 \times 2$ (dof=1) | 0.43 | 0.3 | $0.27 \pm 0.09$ | 10 | 0.071 |
| Student-t $2 \times 2$ (dof=2) | 0.19 | 0.2 | $0.20 \pm 0.02$ | 9 | 0.054 |
| Student-t $3 \times 3$ (dof=2) | 0.29 | 0.2 | $\mathbf{0.26 \pm 0.04}$ | 10 | 0.031 |
| Student-t $3 \times 3$ (dof=3) | 0.18 | 0.1 | $\mathbf{0.18 \pm 0.03}$ | 9 | 0.021 |
| Student-t $5 \times 5$ (dof=2) | 0.45 | 0.4 | $0.37 \pm 0.06$ | 10 | 0.052 |
| Student-t $5 \times 5$ (dof=3) | 0.3 | 0.2 | $0.23 \pm 0.04$ | 10 | 0.054 |
| Swiss roll $2 \times 1$ | 0.41 | 0.4 | $0.38 \pm 0.02$ | 6 | 0.048 |
| Wiggly @ Bivariate normal $1 \times 1$ | 0.41 | 0.4 | $0.39 \pm 0.01$ | 5 | 0.184 |

Table 3: **Improved results on spiral datasets with larger critics.** For challenging cases where smaller critics underperform, increasing the hidden layer width from 16 to 256 substantially improves accuracy. Grey rows indicate unreliable fits.

| Task | True MI | simple InfoNCE | Ours (16 hidden) | Ours (256 hidden) | $\gamma_{max}$ | $\delta$ |
|---|---|---|---|---|---|---|
| Multinormal $25 \times 25$ | 1.02 | 0.7 | $0.97 \pm 0.24$ | $\mathbf{1.11 \pm 0.17}$ | 10 | 0.076 |
| Multinormal $3 \times 3$ | 1.02 | 0.6 | $0.69 \pm 0.04$ | $\mathbf{0.99 \pm 0.05}$ | 10 | 0.061 |
| Multinormal $5 \times 5$ | 1.02 | 0.6 | $0.64 \pm 0.04$ | $\mathbf{1.04 \pm 0.08}$ | 5 | 0.136 |
| Normal CDF @ Multinormal $25 \times 25$ | 1.02 | 0.8 | $0.89 \pm 0.10$ | $\mathbf{0.93 \pm 0.10}$ | 10 | 0.067 |
| Normal CDF @ Multinormal $3 \times 3$ | 1.02 | 0.9 | $0.89 \pm 0.03$ | $\mathbf{1.00 \pm 0.04}$ | 10 | 0.005 |
| Normal CDF @ Multinormal $5 \times 5$ | 1.02 | 0.9 | $0.88 \pm 0.04$ | $\mathbf{0.99 \pm 0.03}$ | 10 | 0.076 |

## A.8 Additional Figures

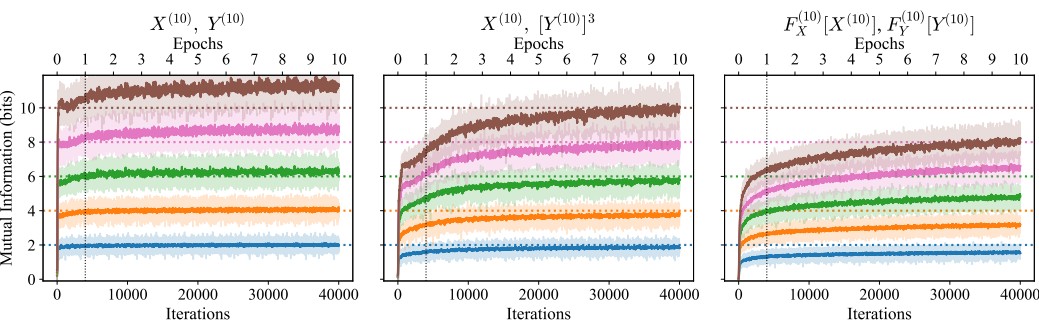

Figure 10: **Effect of continued training on MI estimation in the low-dimensional setting.** Using the SMILE estimator (with a deterministic critic), we replicate the experiment from Fig. 1, but train for 10 epochs instead of just one, revisiting each training sample multiple times. In the raw Gaussian case (left), SMILE begins to significantly overestimate mutual information at high MI levels, consistent with its known overfitting behavior. For the $Y^3$ nonlinearly transformed case (middle), the estimator saturates to the correct MI level only after multiple epochs, suggesting that underestimation in Fig. 1 was due to insufficient training. The teacher network case (right) shows modest improvement with more training but still falls short of the true MI, reflecting the partial information loss introduced by projecting into a 10D space via non-invertible embeddings. This figure underscores the importance of the training regime and data reuse in MI estimation. While fresh batches avoid overfitting, multiple epochs can be critical for extracting information from nonlinear transformations.

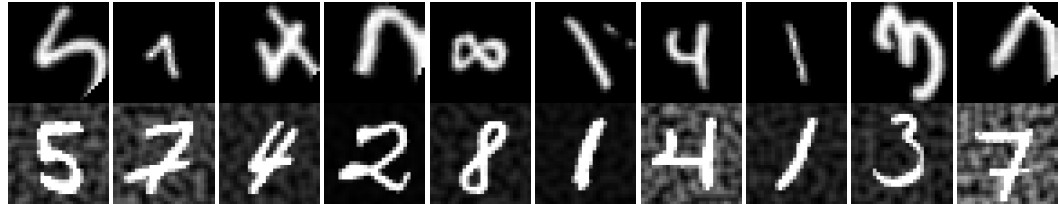

Figure 11: **Sample pairs from the Noisy MNIST dataset.** Each pair $(X, Y)$ shares the same digit class label but consists of distinct, non-overlapping digit instances. The $X$ image is generated by applying a random rotation (uniform between 0 and $\pi/2$) and a random scaling (uniform between 0.5 and 1.5) to an MNIST digit. The $Y$ image is formed by applying a new instant of Perlin noise background to each digit, with the noise weight uniformly drawn from $[0, 1]$. Both images are normalized to the $[0, 1)$ intensity range and flattened to $784$-dimensional vectors. We generate up to $2^{18}$ unique training pairs from the MNIST training set and $1024$ from the test set, ensuring an approximately uniform digit distribution. These data preserve only the class-level semantic information ($I(X; Y) \approx \log_2 10$ bits), providing a high-dimensional test case for mutual information estimation.

## A.9 Use of LLMs.

We used large language models (LLMs) to assist with polishing the writing of the manuscript and with some coding tasks, such as writing helper functions for data parsing and streamlining workflows. All conceptual development, theoretical analysis, experimental design, and result interpretation were conducted by the authors.

