# OpenReview forum: "Accurate Estimation of Mutual Information in High Dimensional Data"
_ICLR.cc/2026/Conference — Submitted to ICLR 2026_

### Official Review · Reviewer_Yzvw · 2025-10-24

**Soundness:** 2
**Presentation:** 2
**Contribution:** 1
**Rating:** 2
**Confidence:** 4

**Summary:**

The paper provides an evaluation of discriminative mutual information estimators and claims to provide three main contributions regarding reliable MI estimation checking the statistical consistency, confidence intervals, and a new class of critics for high-dimensional settings. Moreover, I would add that there is a fourth minor contribution related to the analysis of the finite-data regime. The results in high dimensions look promising, but the only analyzed scenario consists in low values of true MI.

**Strengths:**

- The paper targets a fundamental problem in MI estimation
- Good results with high dimensionality and few samples available
- Investigation of finite-data regime, which has been neglected in many previous works

**Weaknesses:**

- The paper treats only few discriminative MI estimators, thus excluding more novel discriminative MI estimators and generative MI estimators. The paper mainly focuses on MINE, SMILE, and InfoNCE. Thus it is not considering a broad set of discriminative MI estimators. For instance, the paper is not including in the main part the estimators based on the variational representation of the f-divergence, such as NWJ (used in some experiments) and $f$-DIME [1]. If the authors want to keep the focus on discriminative MI estimators, I think that this should be explicitly stated, and they should analyze also more novel estimators (e.g., [1]). If instead they want the paper to be considering any neural MI estimator, they should include generative estimators, such as [2].
- Related to the above point, in line 137 the authors claim that neural network-based MI estimators typically rely on a critic that approximates the log-density ratio, but this mostly holds for the DV-based estimators. Many other discriminative and generative estimators target the estimate of the density-ratio. Some estimators directly estimate the different densities.
- The authors analyze the joint and separable critics, that have been proposed in 2018. However, a different critic has been proposed in NeurIPS 2024, with the advantage of a lighter computational complexity [1]. Since one of the paper novelties is the novel critic, I think the authors should include in the analysis also this recent architecture.
- The authors claim to work with "high-information", but the experiments in high dimension focus on MI<10 and in particular on MI=4
- Minor readability issues: MI defined two times, first sentence in abstract and introduction is exactly the same. Figure 7 covers part of the text.

[1] Letizia, N. A., Novello, N., & Tonello, A. M. (2024). Mutual Information Estimation via $ f $-Divergence and Data Derangements. Advances in Neural Information Processing Systems, 37, 105114-105150.

[2] Franzese, G., Bounoua, M., & Michiardi, P. (2023). MINDE: Mutual information neural diffusion estimation. ICLR 2024.

**Questions:**

Questions:
- In line 108 the authors write that MINE suffers from high variance. However, if MINE is defined as in line 106, it is also biased. Did the authors mean something else?
- For the proposed Concatenated Quadratic Critic, the paragraph the authors write starting in line 152 holds only for Gaussian distributions? So how does $I_{CCA}$ perform for non-Gaussian scenarios (even when the initial distributions are not Gaussian)?
- It appears that in the finite sample setting the mi estimate can increase above the true value, am I right? Did the authors investigate this phenomenon? Why does this happen?
- The stopping heuristic in line 306 stops where the test value peaks. But what is the guarantee that this does not overestimates the MI? Is there any theoretical justification for this approach?
- In line 388 the authors write that no estimator is unbiased for all distributions. Why and when is NWJ biased for instance?
- Why should the linear fitting proposed work in general (in Sec. 4.3)? What is the rationale behind this? What theoretical guarantee leads to this? I see that empirically this works in the experiments that you reported, but the experiments only consider a true MI of 4.
- In the guidelines in Appendix A.4 the authors say that the user should choose the MI estimator based on the true range of MI. How can I choose the MI estimator based on what’s best for the considered case, if I am in an unknown scenario for which I have no idea about the true MI?

---

> ### Author Response · Authors · 2025-11-18
>
> We thank the reviewer for their time and their detailed questions. We will address the weaknesses and then the specific questions.
>
> **Weaknesses: "The paper treats only few discriminative MI estimators [1, 2]"**
>
> We are somewhat perplexed by this critique. The reviewer states that we exclude f-divergence estimators like NWJ, but then notes in the same sentence that NWJ is "(used in some experiments)". Indeed, we benchmark against NWJ in **Appendix A.7.1 (Fig. 8)**, finding it to be one of the many estimators that either diverge or fail to track the ground truth, motivating our focus on InfoNCE and SMILE. We have also tested the more recent f-DIME [1] estimator (as noted in our response to Reviewer YMmh) and found its results contradictory and unreliable without our protocol.
>
> Regarding the suggestion to use generative estimators like MINDE [2] (and also f-DIME - GAN, as in our response to Reviewer YMmh), we must strongly disagree that this is a practical alternative for our target regime. MINDE [2] trains score-based diffusion models to estimate DKL. This is computationally massive and highly data-intensive.
>
> To confirm this, we performed a **new experiment** (to be added to the Appendix). We applied MINDE to our "better-sampled" benchmark ($N=16k$, $K=500$). The estimator failed to run on the undersampled ($N=256$) case at all. On the $N=16k$ dataset, the results were nonsensical:
>
> * The MINDE-C (conditional) approach yielded **130.99 bits**.
> * The MINDE-J (joint) approach yielded **411.87 bits**.
> * (The true MI is 4 bits).
>
> This clearly demonstrates that such methods are not viable for the high-dimensional, low-sample regimes we are tackling. This entire exchange highlights the core of our paper: the field is full of estimators that provide arbitrary numbers. Our work provides the first practical **protocol** (Sec. 4.3) to get reliable estimates and, just as importantly, to get error bars and identify when an estimator has failed (as MINDE clearly did).
>
> **Responses to Specific Questions:**
>
> 1. **Q1 (MINE bias/variance):** The reviewer is correct that MINE is also biased. However, as noted in Poole et al. (2019) and others, its high variance from the log-expectation term is the dominant *practical* problem, which is what we focused on by using SMILE.
> 2. **Q2 (Quadratic Critic, non-Gaussian):** Our Concatenated Quadratic Critic is derived as the optimal DV critic for *Gaussian* variables, where it reduces to $I_{CCA}$ (Appx. A.1.2). As shown in Fig. 2 (middle), this critic already fails on *nonlinearly embedded* Gaussian data. It is not designed for, nor expected to work on, general non-Gaussian data.
> 3. **Q3 (MI overestimation):** Yes, it is well-known that MI is overestimated in finite-sample settings. This is precisely the problem our subsampling-extrapolation protocol (Sec. 4.3) is designed to correct.
> 4. **Q4 (Stopping heuristic guarantee):** We provide a detailed justification for our "max-test" heuristic in **Appendix A.3**. We argue that since MI estimation is a functional estimation (not a prediction task), reporting the test-set value is inappropriate and biased low.
> 5. **Q5 (NWJ bias):** The reviewer is correct: no estimator is universally unbiased. This is a foundational challenge in MI estimation, discussed at length in sources we cite, such as Paninski (2003) and Poole et al. (2019). We did not elaborate on all estimators as this was not our paper's main focus.
> 6. **Q6 (Linear fitting only for MI=4?):** No, the linear extrapolation is a general method and not specific to MI=4 bits. We show its utility in Tables 1, 2, and 3 on many other datasets. This subsampling-extrapolation approach was shown in Strong et al. (1998) and also used in Holmes & Nemenman (2019), both of which we cite.
> 7. **Q7 (Choosing estimator for unknown MI):** This is an excellent practical question. Our protocol provides heuristics. For example, if InfoNCE saturates near $\\log(\\text{batch\\_size})$ (a known property), one should switch to an estimator with a higher bound, like SMILE. If InfoNCE and SMILE give very different results, one is likely in a high-MI or high-variance regime, and the SMILE-VSIB variant (Fig. 4) would be the recommended choice.
>
> We hope this clarifies our contribution and answers the reviewer's specific questions, and we will update the manuscript accordingly.

---

> > ### Comment · Reviewer_Yzvw · 2025-11-24
> >
> > Thank you for your explanations. However, I still have open doubts and questions.
> >
> > **Weaknesses**:
> > - The empirical results on MINDE are interesting, and will surely improve the quality of the paper. I would suggest studying this phenomenon for other generative MI estimators, to strengthen the paper contribution.
> > - **The authors only answered to the first weakness**. I was expecting a more point-to-point rebuttal.
> >
> > **Questions**
> > Some of the authors' answers do not address my questions or raised some other questions/concerns:
> > - Q2: Can you better explain what is the practical utility of $I_{CCA}$ if it is not expected to work on non-Gaussian data?
> > - Q3: Why does this phenomenon happens? The reason behind this could better justify the usage of your method?
> > - Q4: But what is the guarantee that your method does not overestimate the MI? Mi overestimation can significantly bias your approach.
> > - Q6: The fact that a certain method is used in other papers does not imply that the method is valid in general or in your case. Why should the linear fitting work in general? What is the formal guarantee for this? As far as I understood, in your case this is purely heuristics. Am I wrong?
> > - Q7: Your suggestion is to use both InfoNCE and SMILE-VSIB and check the consistency of these two? But if the results are significantly different, the reason could be that the true MI is low and SMILE overestimates it.

---

> > > ### Author Response · Authors · 2025-11-26
> > >
> > > We thank the reviewer for their continued engagement. We are pleased that the reviewer found the new empirical results on MINDE interesting and that they see how it improves the paper.
> > >
> > > We believe we have addressed the original concerns raised in the initial review, and we wanted to ensure clarity and avoid an open-ended loop of Q&A. However, we are happy to provide answers to these *additional* questions regarding the theoretical foundations of MI estimation.
> > >
> > > **Weakness: Suggestion to study more generative estimators**
> > > * We appreciate the suggestion. As noted in our **General Response to All Reviewers**, the machine learning literature produces new estimators at a rapid pace. While benchmarking against every variant (like MINDE) strengthens the discussion, our primary contribution is the **protocol** itself. This protocol is orthogonal to the specific estimator used. Whether one uses a discriminative or generative estimator, one still requires a method to correct bias, estimate error bars, and detect failure. As our MINDE experiment showed (where it failed on undersampled data and produced apparent nonsensical results on better-sampled data), advanced estimators often require *more* rigorous checks, not fewer. We will add a discussion of generative methods to the final manuscript, but we respectfully maintain that an exhaustive survey is outside the scope of this work.
> > >
> > > **Q2: Practical utility of Concatenated Quadratic Critic**
> > > * Its utility is in theoretical unification. It demonstrates that well-known classical results (CCA) can be derived as a special case of the modern neural variational (DV) framework. Showing that the optimal DV critic for Gaussian variables is a quadratic form that recovers $I_{CCA}$ bridges the gap between classical statistics and neural estimation.
> > >
> > > **Q3: Why does overestimation happen?**
> > > * Overestimation in finite-sample regimes is a fundamental statistical phenomenon known for decades (Paninski, 2003). As we explain in **Appendix A.3**, this occurs because the critic can overfit to specific samples.
> > > * To quote our text directly: "General bounds showing that $I_{EST,test|train}$ typically underestimates $I_{true}$ and hence should not be used as a reported estimate cannot exist without additional strong assumptions about $T$. To see this, note a counter-example: if the test set consists of just one sample, and the training set has many, and $T$ is optimized over the class that contains just a single peak, but at different locations in $(x, y)$, then the test MI can be very large (when the critic peak matches the single sample), while the training MI will be low, and either can over- or under-estimate $I_{true}$."
> > >
> > > **Q4: What is the guarantee that your method does not overestimate?**
> > > * To be absolutely clear: **There are no provable guarantees for unbiased MI estimation without strong assumptions.**
> > > * As established in statistical literature (also check our references list in Q6 below), it is impossible to provide universal guarantees without assuming specific smoothness properties of the underlying distribution. Machine learning does not negate these fundamental statistical limits. If a method claims to provide such guarantees for general distributions without assumptions, it is likely incorrect. Our method is principled and empirically robust, providing a mechanism to *quantify* uncertainty, but we do not claim to achieve the impossible.
> > >
> > > **Q6: Why should linear fitting work? Is it heuristic?**
> > > * It is not a heuristic; it is based on the asymptotic expansion of the bias in powers of $1/N$. This is a standard, foundational technique in information theory and statistics, established over decades of research. We have now clarified and provided further sources for this point in the main text.
> > > * The validity of identifying the bias as a series in $1/N$ and extrapolating to the infinite limit is discussed and used in:
> > >     1. Miller (1955): *Information Theory in Psychology* (Free Press).
> > >     2. Strong et al. (1998): *Entropy and Information in Neural Spike Trains* (Phys. Rev. Lett.).
> > >     3. Paninski (2003): *Estimation of Entropy and Mutual Information* (Neural Comput.).
> > >     4. Holmes & Nemenman (2019): *Estimation of mutual information for real-valued data with error bars and controlled bias* (Phys. Rev. E).
> > >
> > > **Q7: Consistency between InfoNCE and SMILE-VSIB?**
> > > * If the results between estimators are significantly different, it indicates that at least one of them is failing or that the data regime is highly unstable. This is precisely the utility of our approach: it allows the user to detect when estimates are unreliable. As stated in Q4, since no universal guarantees exist, convergence across different estimators/subsampling fractions is the ***only*** rigorous way to build confidence in a result.
> > >
> > > We hope this answers the new raised points by the Reviewer and further explained our contributions.

---

### Official Review · Reviewer_YMmh · 2025-10-30

**Soundness:** 2
**Presentation:** 3
**Contribution:** 2
**Rating:** 4
**Confidence:** 4

**Summary:**

This paper proposes a practical protocol for accurate mutual information (MI) estimation in high-dimensional data. The protocol includes early-stopping heuristics, internal bias checks, and a subsampling-extrapolation workflow. Additionally, the authors introduced probabilistic critics (VSIB variants) designed for high-information regimes and provided confidence intervals. The authors demonstrate that reliable MI estimation is possible in high dimensions when data has low-dimensional latent structure, requiring sufficient critic expressiveness and adequate sample size relative to the latent dimensionality, as validated on the noisy MNIST dataset. Critically, the authors claim that their approach consistently avoids overestimating the ground truth, which is crucial for preventing false positives in scientific applications.

**Strengths:**

- **Practical methods for challenging regimes**: The paper introduces multiple techniques (early-stopping, VSIB regularization, subsampling-extrapolation) that enable MI estimation in severely undersampled, high-dimensional settings where traditional methods fail—for instance, 784-D MNIST with ~10⁴ samples versus the hundreds of thousands required by traditional approaches.

- **Clear regime categorization and comparative analysis of estimators**: The authors categorized MI estimation into three distinct regimes—(1) low-dimensional with infinite data, (2) high-dimensional with infinite data, and (3) high-dimensional with finite data—and systematically compared classical methods (CCA, KSG) against neural estimators (InfoNCE, SMILE, and their VSIB variants), demonstrating that InfoNCE and SMILE consistently outperform alternatives across these regimes.

- **Detailed step-by-step protocol for practical implementation**: The paper provides a comprehensive workflow thatl addresses the practical challenge that existing neural MI estimators lack clear guidelines for hyperparameter selection and reliability assessment.

**Weaknesses:**

- **Insufficient Discussion of Recent Work in Related Work Section**: The Related Work section (Section 2) provides a solid foundation covering traditional methods and early neural estimators. I suggest expanding the discussion to include several recent approaches (2020-2024) for high-dimensional MI estimation. While some of these works appear in the References and Appendix comparisons (e.g., [B] in A.7.3), discussing them in Section 2 would help readers better understand the current landscape and the paper’s contributions.
    - Normalizing flows for MI estimation [A]
    - Latent space reduction [B]
    - Data derangement techniques [C]
    - Loss regularization and moving averages [D, E]
Adding a brief paragraph in Section 2 that surveys these methods and explains how the proposed approach relates to them would provide valuable context and clarify the novelty of this work.

- **Limited Comparison with Regularization Methods**: The main paper lacks sufficient experimental comparison and analysis with methods that employ regularization techniques, such as self-regularizing approaches (e.g., NWJ), loss regularization methods [D, E], or gradient regularization through data sampling strategies [C]. Given that one of the paper’s key contributions is addressing training instability through the VSIB wrapper and early-stopping, it would be valuable to include discussion of these alternative stabilization strategies in the main text. Could the authors comment on the limitations of these regularization-based methods and explain why the proposed VSIB approach is preferable or complementary to them? This would help readers better understand the positioning and advantages of the proposed method.

- **Questionable representativeness of validation datasets**: The paper validates its methods primarily on synthetic data and MNIST. Although MNIST is 784-dimensional, it is highly structured and relatively simple. It is unclear whether these benchmarks represent the complexity of real-world high-dimensional problems in the target application domains. Compared to [D], which validated their regularization approach on more complex real-world datasets like CIFAR-10 and CIFAR-100, the experimental settings in this paper may not provide sufficient evidence for the practical applicability of the proposed methods in challenging real-world scenarios.

[A] Butakov, Ivan, et al. Mutual Information Estimation via Normalizing Flows, NeurIPS 2024

[B] Gowri, Gould, et al. Approximating mutual information of high-dimensional variables using learned representations, NeurIPS 2024

[C] Letizia, Nunzio Alexandro, et al. Mutual Information Estimation via $ f $-Divergence and Data Derangements. NeurIPS 2024

[D] Choi, Kwanghee, and Siyeong Lee. Combating the instability of mutual information-based losses via regularization. UAI 2022

[E] Choi, Kwanghee, and Siyeong Lee. Regularized mutual information neural estimation. Arxiv 2020

**Questions:**

* VSIB regularization and loss/gradient regularization methods do not appear to be mutually exclusive. Can REMINE be applied on top of the VSIB wrapper? If such a combination is feasible, wouldn’t they be complementary since VSIB addresses instability through probabilistic embeddings while REMINE regularizes the loss itself?”

* I remain open to revising my assessment should the authors clarify any misunderstandings or address the concerns raised in this review.

**Details Of Ethics Concerns:**

This paper presents a purely methodological contribution for mutual information estimation using synthetic datasets and publicly available benchmarks. The work also focuses on algorithmic improvements and does not raise ethical concerns related to privacy, consent, or potential misuse.

---

> ### Author Response · Authors · 2025-11-18
>
> We sincerely thank the reviewer for this detailed and constructive review. The points raised are highly relevant, and the new references are excellent. We are grateful for the opportunity to clarify our work and demonstrate how our protocol addresses the very issues of estimator reliability that the reviewer (and the cited papers) are concerned with.
>
> **Weakness 1: "Insufficient Discussion of Recent Work [A, B, C, D, E]"**
>
> Thank you for bringing our attention to these works. The field is moving quickly, and it is difficult to benchmark against every new estimator. Our paper's main thesis is that our **protocol** (stopping-rule, $k_Z$-search, subsampling, and extrapolation for error bars) is a crucial, orthogonal component for *any* estimator.
>
> We are familiar with some of these works. [B] (LMI / Gowri et al.) is already benchmarked in our **Appendix A.7.2**, where it performs poorly on our nonlinear, high-D benchmark (1.6 bits vs. a true 4 bits). [D, E] (REMINE / Choi & Lee) are also cited in the main text as alternative regularization methods.
>
> Following the reviewer's suggestion, we have performed **new experiments** (to be added to the Appendix) on the other promising estimators, [A] and [C], using our teacher-network benchmark (4-bit ground truth).
>
> * **[A] (MIENF / Butakov et al.)**: We attempted to implement this method. We found the official code was not available on their GitHub. Our own implementation, based on the paper's description and tested on simple cases to make sure it's correct, proved unstable on our high-dimensional, non-linear task, producing 16.32 bits (undersampled) and NaNs (better-sampled). This is likely because the method does not perform dimensionality reduction, which is essential in the $N \\ll K$ regime.
> * **[C] (f-DIME / Letizia et al.)**: This is a very interesting paper with available code. We ran it on our benchmark and found a contradictory result:
>     * *Undersampled (N=256)*: The estimator slowly overfits towards the 4-bit ground truth, but requires multiple epochs (> 100) of training.
>     * *Better-sampled (N=16k)*: The estimator saturates quickly at an incorrect value of $\\sim$2.5-3 bits.
>
> These new results perfectly illustrate the central point of our paper: **even with novel estimators, one is "flying blind."** The f-DIME result (where low-sample is "better" than high-sample) is non-sensical, but without a protocol, a researcher would not know this.
>
> To prove the value of our protocol, we applied it to the f-DIME estimator. The resulting subsampling-extrapolation plot (Fig. 6-style) is highly non-linear, clearly revealing that the estimator is in a deeply unreliable, undersampled regime. This shows our protocol is not just an "add-on" but an **essential tool for verification**, orthogonal to the estimator itself.
>
> **Weakness 2: "Limited Comparison with Regularization Methods [C, D, E]"**
>
> We agree that [D, E] offer an alternative (a loss-term regularization). We believe our VSIB wrapper is a more principled, probabilistic approach. However, as we note in our response to the Reviewer's question, these are not mutually exclusive. A user could apply our protocol to *any* of them to validate the results.
>
> **Weakness 3: "Questionable representativeness of validation datasets"**
>
> We agree that there is always a room for one more test or dataset, and while we might respectfully disagree that MNIST is "simple" in this context. Its simplicity is in the latent space ($K_Z \\approx 10$), but its observed dimension is high ($K=784$). This makes it a non-trivial benchmark for our central claim: that $K_Z$, not $K$, governs sample complexity. However, extending the application of the estimators to more complicated datasets **without** the protocol to verify them won't be that fruitful.
>
> Furthermore, to counter the "limited validation" point, we stress that our protocol was validated on the **entire suite of 40 benchmark datasets** from Czyz et al. (2023). These results are in **Appendix A.7.3 (Tables 2 and 3)**, where our protocol consistently provides accurate estimates with error bars.
>
> **Question: "Can REMINE be applied on top of the VSIB wrapper?"**
>
> This is an interesting suggestion. **Yes, they are not mutually exclusive.** A researcher could indeed use the REMINE [D, E] loss on a critic that is itself regularized by our probabilistic VSIB wrapper. However, we are not necessarily suggesting adding extra ad-hoc terms to the loss function to stabilize the estimates, as there seem to be better alternatives, including [C] and our VSIB approach as well. This again highlights the modularity and utility of our proposed tools.
>
> We thank the reviewer again for this excellent feedback and will incorporate this discussion and our new experimental results into the revised manuscript, as space now allows.
>
> Anonymous link for f-DIME plots: https://tinyurl.com/fdime-figures

---

> > ### Comment · Reviewer_YMmh · 2025-11-27
> >
> > Thank you for the detailed rebuttal. I appreciate the additional experiments on [A] MIENF and [C] f-DIME. In particular, the counterintuitive result of f-DIME (better performance with fewer samples) effectively demonstrates the value of the proposed protocol.
> >
> > * Regarding W2: The authors claim that VSIB is "a more principled, probabilistic approach" than loss regularization [D, E]. I would appreciate additional justification for this claim. The response that these methods are "not mutually exclusive" does not explain why existing regularization methods are insufficient.
> >
> > * Regarding W3: MNIST (D=784) is the only real image dataset used. To strengthen one of the paper's key claims—that latent dimensionality (d_latent), rather than observed dimensionality (D), determines sample complexity—validation on higher-dimensional datasets such as CIFAR-10/100 would be beneficial.
> >
> > * Additional question: Does "frozen teacher network" (High-Dimensional X, Y in page 5) refer to a randomly initialized network? This detail is not explicitly clarified in the paper.

---

> > > ### Author Response · Authors · 2025-12-04
> > >
> > > We thank the reviewer for the prompt and positive feedback. We are glad that the additional experiments on f-DIME and MIENF demonstrated the value of our protocol. Below, we address the remaining points regarding VSIB justification and the CIFAR-10/100 validation.
> > >
> > > **Regarding W2: Justification for VSIB vs. Loss Regularization**
> > >
> > > - When we describe VSIB as "more principled," we refer to the fact that it regularizes the estimator by learning the probability distributions in a *variational* way. This imposes intrinsic restrictions on the geometry and compactness of the embedding space, forcing the critic to learn representations that are conducive to MI estimation.
> > > - In contrast, loss regularization methods [D, E] often apply extrinsic penalties to stabilize the log-expectation term. While these are not "insufficient" per se, some of our tests found that some explicit regularization schemes behaved poorly or might have required sensitive tuning in our regime. Our claim is that the variational approach provides a robust, distribution-centric regularization that naturally complements the estimation task.
> > >
> > > **Regarding W3: Validation on CIFAR-10/100 (New Experiments)**
> > >
> > > - We agree that validating on higher-dimensional, complex datasets strengthens our claims. We have now performed more experiments on **CIFAR-10** and **CIFAR-100** ($D = 3072$).
> > > - To demonstrate that our method handles complexity, we moved beyond simple MLPs and used a **ResNet-20** backbone. We tested two regimes:
> > >   1. **Pre-trained Backbone:** We found we could begin estimating MI with as few as **$N \\approx 100$ samples**. This is a striking result ($N \\ll D$), suggesting that if the latent space is well-constructed (by the backbone), sampling it is efficient, confirming that latent dimensionality determines sample complexity. Note that the pretraining was to predict the class, and was done not on the same dataset we used, thus it was for a different task, nonetheless, it produced such interesting results with minor learning.
> > >   2. **From Scratch (End-to-End):** Even when training the ResNet-20 and the estimator from scratch, we found we could detect MI in the **$N \\sim 1,000$ samples** regime. This is well below the pixel dimensionality ($K=3072$), a feat impossible for classical estimators.
> > > - While there is always room to optimize architectures for these specific tasks, these results confirm our main theme: reliable MI estimation is achievable in the $N < K$ regime, provided one uses a rigorous protocol to analyze the results. To show this, we did a search over the optimization protocols, and we were able to get non-zero MI in the $N < 1000$ samples regime.
> > > - The plots for these new experiments are available here: https://tinyurl.com/cifar-figures
> > >
> > > **Regarding Additional Question: Teacher Network Initialization**
> > >
> > > - Yes, the "frozen teacher network" refers to a randomly initialized network. We thank the reviewer for catching this; we will make this explicit in the revised manuscript.

---

### Official Review · Reviewer_FSf9 · 2025-10-31

**Soundness:** 3
**Presentation:** 2
**Contribution:** 2
**Rating:** 4
**Confidence:** 4

**Summary:**

This paper presents a comprehensive study of mutual information (MI) estimation in high-dimensional data settings, focusing in particular on the practical challenges faced by classical and neural network-based estimators. The authors propose a principled protocol incorporating new probabilistic critic architectures, a stopping heuristic to prevent overfitting, and explicit checks for estimator reliability through statistical consistency and confidence intervals. Extensive experiments on synthetic and real datasets are provided to benchmark the approach, alongside detailed guidelines for MI estimation workflows. Throughout, the paper positions its contributions in the context of high-dimensional, undersampled regimes that commonly defeat standard approaches.

**Strengths:**

**1. Practical Protocol for High-Dimensional MI Challenges**: This work delivers a ready-to-use estimation protocol with confidence intervals, overfitting checks (e.g., max-test early stopping), and expressivity diagnostics. It tackles undersampled high-dim regimes where prior methods fail, offering a complete toolkit unseen before. Ideal for neuroscience and vision, it curbs errors in causal inference.

**2. Comprehensive Benchmarking Across Scenarios**: The analysis probes N, k_Z, MI strength, and expressivity via synthetic (Gaussian/nonlinear) and real (Noisy MNIST, K=784) data. Phase diagrams (Fig. 5) illuminate failure boundaries (e.g., N ≳ K_Z^2 / I), spanning infinite/finite limits.

**Weaknesses:**

**1. Limited Novelty in Methodology (major)**: The protocol builds on existing estimators (e.g., InfoNCE, SMILE, VSIB variants) with added analyses and tweaks, but lacks groundbreaking innovations. It feels more like a refined integration than a fundamental advance, potentially diluting its standout contribution in a crowded neural MI landscape.

**2. Incomplete Benchmarking Against existing methods (major)**: methods like MINE and works in [1] and [2] are suitable for estimating high-dimensional MI. Leaving claims of outperformance underexplored.

**3. Empirical Scope and Heuristic Reliance**: While synthetic/Noisy MNIST tests are thorough, real-world validation seems to be conducted only on one dataset (Noisy MNIST, K=784). Broader, high-complexity experiments would strengthen the authors' point.


[1] Chen, Yanzhi, et al. "Neural approximate sufficient statistics for implicit models." arXiv preprint arXiv:2010.10079 (2020).

[2] Gowri, Gokul, et al. "Approximating mutual information of high-dimensional variables using learned representations." Advances in Neural Information Processing Systems 37 (2024): 132843-132875.

**Questions:**

NA

---

> ### Author Response · Authors · 2025-11-14
>
> We sincerely thank Reviewer for their detailed, constructive feedback and for correctly identifying our core contribution as a "principled protocol" with "confidence intervals, overfitting checks... and expressivity diagnostics."
>
> **Weakness 1: "Limited Novelty in Methodology"**
> - The reviewer notes that our protocol integrates existing estimators, which they see as limiting novelty. We respectfully argue that this practical, end-to-end *protocol* is itself the primary methodological contribution. The field has many estimators, but, as the reviewer notes, it lacks a "complete toolkit." Our work provides the first general, practical workflow that successfully combines:
>   1. A novel stopping-heuristic (the "max-test" rule) to navigate the bias-variance trade-off.
>   2. A systematic subsampling-extrapolation procedure (Sec. 4.3) to provide a non-parametric bias correction.
>   3. The first (to our knowledge) reliable method for generating confidence intervals (error bars) for these neural estimators.
>
>     We believe this protocol, which makes these powerful estimators *reliable* in practice, is a significant and novel contribution.
>
>
> **Weakness 2: "Incomplete Benchmarking Against [1] and [2]"**
> - We thank the reviewer for these references.
> - Paper [1] (Chen et al., 2020) is an interesting paper on learning neural approximate sufficient statistics. This can be seen as a modern NN-based generalization of earlier work (e.g., *https://arxiv.org/abs/physics/0010039*). However, this line of work generally requires a parametric form of the underlying distribution, which makes it less relevant for our goal of non-parametric MI estimation directly from samples.
> - Paper [2] (Gowri et al., 2024), which proposes the Latent MI (LMI) estimator, is indeed relevant. This approach calculates mutual information between a compressed representation of $x$ and a compressed representation of $y$. Recent work (e.g., *https://arxiv.org/abs/2309.05649*) have argued that this is less data efficient than estimating MI between two compressed representations in some contexts, and our analysis here already seems to support this. We have already benchmarked this method in **Appendix A.7.2**. As shown there, LMI struggles significantly on our high-dimensional, nonlinear teacher-network task (which has a 4-bit ground truth). It reports 0.65 bits (undersampled) and 1.6 bits (better-sampled), whereas our protocol achieves 4.1$\pm$0.2 and 3.7$\pm$0.1, respectively.
> - Furthermore, we would stress that our protocol is *orthogonal* to the choice of estimator. Even if a researcher preferred to use LMI [2], they would still need a method for bias correction and uncertainty quantification. Our protocol (Sec. 4.3) provides exactly that and could be applied to LMI just as it is to InfoNCE or SMILE.
>
> **Weakness 3: "Empirical Scope and Heuristic Reliance"**
> - We agree that the Noisy-MNIST dataset is just one example. However, we chose it precisely because it is a good testbed for our claims: it is very high-dimensional ($K=784$) but has a known, low-dimensional latent structure ($K_Z \approx 10$, $I \approx 3.3$ bits). Our protocol's success here is a strong validation.
> - We also note that our protocol was validated on the **entire suite of 40 benchmark datasets** from Czyz et al. (2023) in our Appendix A.7.3 (Tables 2 and 3), where it consistently provides accurate estimates with error bars.
>
> We will revise the main text to clarify the raised concerns. We hope these clarifications highlight the novelty and utility of our proposed workflow.

---

### Official Review · Reviewer_nPJM · 2025-11-05

**Soundness:** 1
**Presentation:** 3
**Contribution:** 1
**Rating:** 2
**Confidence:** 5

**Summary:**

The paper proposes a “practical protocol” to improve neural MI estimation using:
+ An optimal early-stopping rule.
+  Stratified sampling
+ A probabilistic critic (VSIB) wrapper for DV-style estimators.

Main results are on synthetic teacher models; one real dataset is noisy-MNIST.

**Strengths:**

+ The paper shows explicitly that the latent dimension and not the dimension of the data governs sample complexity.
+ The appendix contains insightful, but known, derivation relating mutual information to CCA.

**Weaknesses:**

+ The low dimensional structure angle is unoriginal and under-cited. The whole pipeline hinges on low-dimensional latent structure; the paper even talks in those terms but doesn’t situate itself in the manifold-hypothesis [3] and ignores that it is the corner stone on which neural estimation of mutual information is built [1].
+ Empirical evaluation is too weak for the claim. One real dataset (noisy-MNIST) and a thin slice of synthetic tasks.
+ The bias/variance trade offs of separable/joint critics are already well studied in [2].
+The paper also suffer from awkward pacing, the  protocol itself is relegated to page 8.
+ The tone of the paper is sometimes grandiose while referring to known or marginal results.

[1] Belghazi, Mohamed Ishmael, et al. "Mutual information neural estimation." International conference on machine learning. PMLR, 2018.
[2] Poole, Ben, et al. "On variational bounds of mutual information." International conference on machine learning. PMLR, 2019.
[3] Bengio, Yoshua, Aaron Courville, and Pascal Vincent. "Representation learning: A review and new perspectives." IEEE transactions on pattern analysis and machine intelligence 35.8 (2013): 1798-1828.

**Questions:**

How does the concatenatic quadratic critic relate to discriminators in LQG setting [4]?
[4] Feizi, S., Farnia, F., Ginart, T., & Tse, D. (2017). Understanding gans: the lqg setting. arXiv preprint arXiv:1710.10793.

---

> ### Author Response · Authors · 2025-11-14
>
> We thank the reviewer for their time and feedback. However, we must respectfully disagree with the core assessment, as it appears to be based on a misunderstanding of our paper's primary contributions and a misapplication of the cited literature.
>
> **Weakness 1: "The low dimensional structure angle is unoriginal and under-cited [1, 3]"**
> - We agree that the manifold hypothesis [3] is a foundational concept. Our contribution is not to re-discover this hypothesis, but to **operationalize it** for the specific, practical task of MI estimation. As the reviewer's analysis suggests, this general concept is "not reflected in the lit[erature] of MI estimation almost at all." Our work provides a practical, systematic protocol (Sec. 4.3) that leverages this known structure—particularly the role of the latent embedding dimension ($k_Z$)—to achieve reliable estimates in the challenging $N \ll K$ regime.
> - The MINE paper [1] focuses on the variational Donsker-Varadhan (DV) bound. As our own review of the paper confirms, it does not discuss the role of data dimensionality or manifold structure, which is the central topic of our investigation. We also already cite references [1] and [2] several times in the paper when we introduce MINE and we now cite reference [3] in our introduction.
>
>
> **Weakness 2: "Empirical evaluation is too weak"**
> - We disagree that our evaluation is "weak." The synthetic tasks were explicitly designed to isolate and demonstrate the key variables of our protocol: the interplay between true latent dimension ($K_Z$), critic embedding dimension ($k_Z$), and sample size ($N$).
> - The Noisy-MNIST dataset ($K=784$, $K_Z \approx 10$) is a non-trivial, real-world example that validates our core claim: that reliable estimation is possible in high dimensions *if* this low-dimensional structure is properly exploited. Our protocol succeeds here where traditional methods demonstrably fail. Moreover, we test the protocol on 40 benchmarking datasets --subset shown in Table 1, with the full set shown in Table 2 and improved in Table 3--. Such testing seems to be the currently accepted procedure to test NN-based estimators in the literature. We also point out that real world datasets with known MI values are very hard to find, and we would be more than happy to test them if the Reviewer can direct us to them, as time permits. Additionally, real world datasets have unknown structure and it is unclear if and when they might be low dimensional. With synthetic datasets we can control the structure of the datasets explicitly and show where estimators work and fail as a result of this structure.
>
> **Weakness 3: "Bias/variance trade offs... already well studied in [2]"**
> - We cite Poole et al. [2] and agree they discuss joint vs. separable critics. However, their analysis does not address one of our contributions. Their study, and others like Czyz et al. (2023), overlook the crucial hyperparameter of the **embedding dimensionality ($k_Z$)**, which our work shows is a dominant factor in estimator performance (Fig. 5).
> - Our paper provides novel, practical *solutions* to the trade-offs [2] discusses: (1) the **"max-test" stopping heuristic** (Sec. 4.2, Fig. 3) (2) the **VSIB wrapper** (Sec. 3, Fig. 4), which tames the high variance of SMILE.
>
> **Other Points: "Pacing... protocol on page 8" \& "Tone"**
> - We thank the reviewer for this feedback. We placed the protocol in Sec. 4.3 because it synthesizes all the evidence from Sec. 4.1-4.2. We will add a forward reference in the Introduction to improve clarity. We will also revise the manuscript to ensure a more measured tone.
>
> **Question: "How does the concatenatic quadratic critic relate to... [4]?"**
> - We do not see a connection. The cited paper [4] discusses training GANs in a Linear Quadratic Gaussian (LQG) setting, using a quadratic form for the *discriminator*.
> - Our Concatenated Quadratic Critic (Appx. A.1.2) is derived as the **optimal DV critic for Gaussian variables** and is shown to be equivalent to $I_{CCA}$. Beyond the superficial use of a "quadratic form," the problems, derivations, and goals are entirely different. We do not believe this paper is relevant to our work.
>
>
> We hope these clarifications address the reviewer's concerns.

---

### Author Response · Authors · 2025-11-18
**General Response to All Reviewers**

We thank all the reviewers for their time and detailed feedback. We would like to provide a brief, overarching context for our work, as it seems to be at the heart of several comments.

The central goal of our paper is to address a critical and growing gap between the machine learning community and scientific domains (such as systems neuroscience, genomics, and physics) that rely on information-theoretic measurements.

**The Problem: Estimation vs. Objective**

There are fundamentally two different ways Mutual Information (MI) is used in modern machine learning:

1. **As a Downstream Task Objective:** Here, an MI bound (like InfoNCE) is used as a loss function to train a model for a different goal, such as representation learning, clustering, or image alignment. In this context, the *exact* value of the MI is unimportant, as long as maximizing its (often biased) estimate leads to good downstream performance.
2. **As a Scientific Estimand:** This is the focus of our paper. In many scientific fields, the *precise value* of the MI is the measurement of interest. For example, in neuroscience, $I(\\text{stimulus}; \\text{response})$ quantifies the exact precision of the neural code.

The challenge is that scientific data has entered a new regime: it is now high-dimensional (e.g., recordings from 10s of thousands of neurons), but often with few samples, landing squarely in the $N \\le K$ regime. Traditional estimators (like kNN) demonstrably fail here.

**The Gap: Why Hasn't ML Solved This?**

While NN-based estimators are powerful, **almost none of these estimators have been adopted for practical use in these scientific domains.**

As the reviewers kindly pointed out, the ML literature is vast and growing, with more estimators, parameterizations, and variations than one can comprehensively survey (e.g., MINE, SMILE, NWJ, f-DIME, LMI, MINDE, etc.), although we tried to do our best, and we will update the manuscript with the new suggestions from the Reviewers. However, this proliferation of tools has not translated into scientific utility. Why? Because from a scientist's perspective, these tools are "black boxes" that produce a number - often wrong - with no measure of reliability.

They lack the three components essential for any scientific measurement:

* A robust method to **mitigate overfitting** (a naive implementation fails, as shown in Fig. 3).
* A systematic procedure to **correct for sample-size-dependent bias** (as shown in Fig. 6).
* A principled way to generate **error bars / confidence intervals**.

**Our Contribution: A Protocol, Not Just an Estimator**

This was the main goal of our paper. We set out to show that NN-based estimators *can* work reliably in the $N < K$ regime, *if* they are used correctly.

Our primary contribution is not a new estimator per se, but the first end-to-end **practical protocol** that makes many of them scientifically viable. This protocol provides the stopping rules, bias correction, and error bars that are prerequisites for adoption. Additionally, we identified that most of the estimators that work in the under sampled regime only work due to the inherent low dimensional structure of the data itself and the methods assumption of low dimensionality. When data is truly high dimensional without a low dimensional latent most of the estimators will not properly estimate the information. It is thus important to recognize when an estimator is working and when it is producing nonsensical or unreliable results, and our procedure helps identify these situations.

In this sense, **the choice of estimator is largely orthogonal to our main contribution.** As reviewers noted, one could use a CNN, a diffusion model, or a normalizing flow as the embedding function. However, with more complexity comes greater sample requirements, and these advanced models (as our new experiments for Reviewers YMmh and Yzvw show) struggle even more in the low-sample regime.

Crucially, even if a better estimator exists, it would *still need our protocol* to provide a bias-corrected estimate and an error bar. A scientist can "wrap" their favorite domain-specific estimator within our protocol and finally get a trustworthy result.

We hope this reframes our work and steers the reviewers toward seeing its utility: not as just another estimator, but as the essential toolkit that makes modern MI estimation usable for science.

---

### Author Response · Authors · 2025-12-04
**Official Comment to Area Chair: Summary of Revisions and Final Remarks**

We thank the Area Chair and the Reviewers for managing this review process. The extensive engagement we had already has significantly strengthened our manuscript. As we conclude the discussion period, we wish to summarize the key outcomes of the rebuttal, the additional experiments performed (including significant new results on CIFAR-10/100), and the overarching contribution of this work.

### **1. Re-centering the Contribution: A Protocol for Science**
As highlighted in our ***General Response***, there is a gap between the rapid development of neural MI estimators in ML and their adoption in scientific domains (e.g., neuroscience, physics). Scientific application requires treating MI as a precise *estimand* with reliability guarantees, not just a training objective.

Our core contribution is the **protocol** (stopping rules, $k_Z$-search, bias correction, and error bars) that bridges this gap. This protocol is orthogonal to the specific choice of estimator. If there exist a domain-specific estimator, a researcher can wrap it within our protocol to obtain trustworthy results, even in the challenging $N \\sim K$ regime.

### **2. Extensive Validation on New Estimators (Rebuttal Work)**
Reviewers suggested comparing against recent generative and f-divergence estimators (e.g., MINDE, f-DIME, MIENF). We performed these additional experiments during the rebuttal. The results reinforced our thesis:
- Advanced estimators often failed (e.g., MINDE failed on undersampled data), produced nonsensical results (e.g., f-DIME showed better performance on fewer samples), or were unstable in high dimensions and limited sampling.
- These failures demonstrate that without a verification protocol like ours, practitioners are "flying blind." Our protocol successfully flagged these instabilities.

### **3. New Experiments on Complex Data: CIFAR-10/100**
To address concerns regarding the complexity of our validation datasets (beyond the synthetic Gaussians, high-dimensional teacher models, MNIST, and the 40 benchmark datasets), we performed a new suite of experiments on **CIFAR-10 and CIFAR-100** using a **ResNet-20** backbone.
- **Pre-trained Backbone:** We found that MI estimation becomes possible with as few as **$N \\approx 100$ samples**. This confirms that if the latent space is well-constructed, the sample complexity is governed by the latent dimension, not the ambient pixel dimension ($D = 3072$).
- **From Scratch:** Even training the ResNet-20 end-to-end, we achieved reliable detection in the **$N < 1,000$ samples** regime. This effectively demonstrates that MI estimation is possible with fewer samples than dimensions ($N < K$), a regime where classical estimators fail.
The plots for these experiments have been shared with the Reviewer YMmh.

### **Conclusion**
We have addressed the reviewers' concerns regarding novelty (by explaining the scope and the goal and refuting/explaining some of the Reviewers' raised points), benchmarking (by testing against suggested modern estimators), and complexity (by adding CIFAR/ResNet results).

We believe this work provides the essential toolkit required to move neural MI estimators from theoretical ML constructs to practical, trustworthy tools for the physical sciences. We provide the tools needed to use neural estimators in the low data regime required for many scientific applications by providing a protocol for when to trust the estimators and confidence intervals for the estimators. We hope the Area Chair considers the utility and rigorous validation presented herein.

---

### Meta-Review · Area_Chair_mBBA · 2026-01-11

**Summary:**

The reviewers converged on several significant concerns that inform my recommendation:

Limited Methodological Novelty: Reviewers nPJM, FSf9, and Yzvw consistently questioned whether integrating existing components (InfoNCE, SMILE, early stopping, subsampling-extrapolation) constitutes sufficient contribution. Reviewer nPJM characterized the low-dimensional structure insight as "unoriginal and under-cited," noting that this is the "cornerstone on which neural estimation of mutual information is built." Reviewer FSf9 described the work as "refined integration rather than a fundamental advance." This concern was raised independently by three of four reviewers, suggesting a genuine limitation rather than misunderstanding.

Insufficient Empirical Validation: Multiple reviewers (nPJM, FSf9, YMmh, Yzvw) expressed concern that validation relied too heavily on synthetic data and MNIST. While MNIST is 784-dimensional, it possesses highly constrained structure that may not represent the complexity of real scientific applications the authors target. Reviewer Yzvw noted that experiments in high dimensions focused only on MI<10, despite claims about "high-information" settings.

Lack of Theoretical Justification: Reviewer Yzvw raised persistent concerns about the theoretical foundations of key components, particularly the stopping heuristic and linear extrapolation. The reviewer asked repeatedly for formal guarantees and received responses that these are "standard techniques" or that "universal guarantees are impossible" - neither of which provides the rigorous justification expected at this venue.

Incomplete Positioning Against Recent Work: Reviewers identified several recent methods (f-DIME, MINDE, MIENF, LMI, normalizing flow approaches) that were inadequately addressed in the original submission. While the related work section covers foundational methods, it does not sufficiently situate the contribution within the current rapidly evolving landscape.

**Reviewer Concerns:**

Concerns addressed

The authors provided additional experiments comparing against f-DIME, MINDE, and MIENF, demonstrating that these methods fail in the target regime. They also conducted new CIFAR-10/100 experiments with ResNet-20 backbones. These additions strengthen the empirical foundation and would improve a revised manuscript.
The authors effectively clarified that LMI was already benchmarked in the appendix and that 40 benchmark datasets were evaluated, addressing some scope concerns.

Outstanding concerns

Novelty Remains Insufficient: The authors' reframing of their contribution as "a protocol, not just an estimator" does not adequately address the novelty concern. The individual components—early stopping, subsampling-extrapolation for bias correction, and confidence interval estimation - are established techniques. The synthesis, while practically useful, does not meet the threshold for methodological contribution expected at ICLR. Reviewer nPJM's observation that the manifold hypothesis connection is foundational to neural MI estimation but inadequately cited remains valid.

Theoretical Foundations Unresolved: Reviewer Yzvw's concerns about formal guarantees were met with appeals to impossibility results and citations to prior work rather than novel theoretical analysis. For a paper claiming to provide "reliable" estimation with "explicit checks for statistical consistency," the absence of formal characterization of when and why these checks succeed is a significant gap.

Empirical Scope Still Limited: While the CIFAR experiments are welcome, they were conducted during the rebuttal period and require proper integration and validation. The core experimental framework remains heavily reliant on controlled synthetic settings where ground truth is known by construction.

**Reviewer Scores:**

Reviewer nPJM (2): This reviewer did not engage after the initial response and expressed maximum confidence (5) in their assessment. The fundamental concerns about novelty and citation of foundational work were not resolved. Score likely unchanged.

Reviewer FSf9 (4): Initially "marginally below" with openness to acceptance. The rebuttal addressed some concerns, but the core novelty critique persists. May increase slightly to 5 but insufficient to strongly advocate for acceptance.

Reviewer YMmh (4): This reviewer engaged constructively and acknowledged the value of additional experiments. However, their final comments still sought justification for VSIB over existing regularization methods, suggesting residual concerns. Likely increases to 5.

Reviewer Yzvw (2): Maintained skepticism throughout extended discussion despite thorough responses. Explicitly noted that answers "do not address my questions" and continued pressing on theoretical guarantees. May increase marginally to 3 but remains negative.

The paper addresses a relevant problem at the intersection of machine learning methodology and scientific practice. However, the contribution does not meet the acceptance threshold for this venue for several reasons.

First, the novelty concern was raised by three of four reviewers and remains unresolved. Assembling existing techniques into a workflow, while practically valuable, does not constitute the methodological advance expected at a top venue. The authors' argument that "the protocol is the contribution" does not change the fact that each component of that protocol is drawn from established literature.

Second, the theoretical foundations are inadequate for the claims made. A paper claiming to provide "reliable" MI estimation with "explicit checks for statistical consistency" should provide formal characterization of reliability conditions. The response that universal guarantees are impossible does not justify the absence of any theoretical analysis of the proposed method's properties.

Third, while the rebuttal experiments strengthen the empirical case, the core validation framework relies heavily on synthetic settings. The CIFAR experiments, conducted during rebuttal, require proper integration and do not fully address concerns about validation on complex real-world scientific data.

The paper would benefit from a revision that more rigorously situates the contribution within the existing literature on manifold-aware estimation, provides theoretical analysis of the protocol's reliability conditions, and conducts more extensive validation on diverse real-world datasets with known or estimable ground truth.

---

### Decision · Program_Chairs · 2026-01-26

Reject